# Uncovering and quantifying the subduction zone sulfur cycle from the slab perspective

Ji-Lei Li [1,2,3]*, Esther M. Schwarzenbach[4], Timm John[4]*, Jay J. Ague[3], Fang Huang [5], Jun Gao[1,2,6]*, Reiner Klemd[7], Martin J. Whitehouse [8] & Xin-Shui Wang[1,2]

Sulfur belongs among $H_2O$, $CO_2$, and Cl as one of the key volatiles in Earth's chemical cycles. High oxygen fugacity, sulfur concentration, and $\delta^{34}S$ values in volcanic arc rocks have been attributed to significant sulfate addition by slab fluids. However, sulfur speciation, flux, and isotope composition in slab-dehydrated fluids remain unclear. Here, we use high-pressure rocks and enclosed veins to provide direct constraints on subduction zone sulfur recycling for a typical oceanic lithosphere. Textural and thermodynamic evidence indicates the predominance of reduced sulfur species in slab fluids; those derived from metasediments, altered oceanic crust, and serpentinite have $\delta^{34}S$ values of approximately −8‰, −1‰, and +8‰, respectively. Mass-balance calculations demonstrate that 6.4% (up to 20% maximum) of total subducted sulfur is released between 30–230 km depth, and the predominant sulfur loss takes place at 70–100 km with a net $\delta^{34}S$ composition of −2.5 ± 3‰. We conclude that modest slab-to-wedge sulfur transport occurs, but that slab-derived fluids provide negligible sulfate to oxidize the sub-arc mantle and cannot deliver $^{34}S$-enriched sulfur to produce the positive $\delta^{34}S$ signature in arc settings. Most sulfur has negative $\delta^{34}S$ and is subducted into the deep mantle, which could cause a long-term increase in the $\delta^{34}S$ of Earth surface reservoirs.

[1] Key Laboratory of Mineral Resources, Institute of Geology and Geophysics, Chinese Academy of Sciences, Beijing 100029, China. [2] Innovation Academy for Earth Science, Chinese Academy of Sciences, Beijing 100029, China. [3] Department of Geology and Geophysics, Yale University, 06520 New Haven, USA. [4] Institut für Geologische Wissenschaften, Freie Universität Berlin, D-12449 Berlin, Germany. [5] Tetherless World Constellation, Rensselaer Polytechnic Institute, Troy 12180 NY, USA. [6] College of Earth and Planetary Sciences, University of Chinese Academy of Sciences, Beijing 100049, China. [7] GeoZentrum Nordbayern, Universität Erlangen–Nürnberg, D-91054 Erlangen, Germany. [8] Department of Geosciences, Swedish Museum of Natural History, SE-104 05 Stockholm, Sweden. *email: lijilei@mail.iggcas.ac.cn; timm.john@fu-berlin.de; gaojun@mail.iggcas.ac.cn

Sulfur is one of the most common volatiles on Earth. It plays key roles in, for example, the redox evolution of the sub-arc mantle[1,2], the formation of ore deposits[2], and the composition of the atmosphere through volcanic $SO_2$ degassing[3]. Subduction zones are the primary locations for the global sulfur cycle, transporting sulfur to the deep mantle via the descending slab or returning it to the surface by arc magmatism[2,4,5]. Compared to fresh MORB, the relatively high sulfur concentrations ([S], up to 3000 μg g$^{-1}$) and positive $\delta^{34}S$ values (+5 to +11‰) of volcanic rocks and melt inclusions in some arcs (e.g., Western Pacific)[4–7], and the presence of sulfate in mantle xenoliths[8], have been attributed to the addition of slab-derived sulfate to arc magmas by fluids[8,9]. Alternatively, some deep arc cumulates (e.g., Eastern Pacific) with mantle-like $\delta^{34}S$ values suggest more limited slab-derived sulfate contributions to arc lavas and that the positive $\delta^{34}S$ signature of the lavas results from crustal assimilation[10].

The role of slab fluids in delivering sulfur species to the mantle wedge is central to this debate. Experimental results suggest that slab-derived aqueous fluids are an effective agent for transporting sulfur from the slab to the mantle wedge[11,12]. In addition, some studies predict that sulfates are likely the dominant sulfur species in slab-derived fluids[9,13]. On the other hand, sulfate is relatively rare in high-pressure (HP) rocks[13–18], and experimental studies have proposed that reduced sulfur species are dominant in slab fluids[11]. Furthermore, in situ measurements of the $\delta^{34}S$ compositions of sulfides from HP eclogites and serpentinites reveal significant isotopic heterogeneity and complicated sulfur behavior during slab metamorphism and metasomatism[13–16].

Clearly, large gaps in our knowledge of the speciation, flux, and isotopic composition of sulfur in slab fluids remain. Understanding these is of utmost importance for addressing slab–arc sulfur recycling, and has global geochemical significance for deciphering the redox state of the mantle[2] and constraining the formation of arc-related ore deposits[12]. Direct examination of devolatilization pathways in exhumed HP rocks is essential to provide independent new perspectives critical to resolving this debate, as it provides the necessary field-based evidence for the sulfur redox state and $\delta^{34}S$ signature of fluids released from subducted slabs.

Sulfur is transported into the subduction zone by sediments, variably altered oceanic crust (AOC), and hydrated slab mantle (serpentinites)[19–22]. Sulfides are commonly observed in exhumed fragments of the oceanic lithosphere such as eclogites, blueschists, HP-metapelites, and serpentinites, as well as related HP veins[13–18,23]. Such vein systems represent fossilized pathways for channelled flow of dehydration-related slab fluids and, thus, directly record fluid geochemical signatures[17,24,25]. Consequently, the study of HP vein–rock systems provides important information

regarding sulfur behavior during slab dehydration and fluid transfer[17]. Isotopic constraints on S-bearing HP rocks and veins linked to the sequence of slab dehydration allow quantification of sulfur release during subduction of oceanic lithosphere.

Here, we report bulk-rock and in situ sulfur isotope compositions for sulfide-bearing HP rocks and veins from the late Paleozoic southwestern Tianshan (ultra-)high-pressure/low-temperature ((U)HP/LT) metamorphic belt (China). The sulfides in these HP rocks and veins[17] provide an exceptional window into the fate of subducted sulfur. Analytical data and thermodynamic calculations point to low sulfur concentrations in slab fluids, which have negative $\delta^{34}S$ values and are dominately composed of reduced sulfur species. Hence, we determine modest slab-to-arc sulfur transport, and find neither significant slab sulfate flux to the mantle wedge nor a direct link between slab-derived sulfur and the positive $\delta^{34}S$ signature of arc settings.

## Results

**Sample background.** The Tianshan (U)HP/LT terrane is an example of deeply buried, uppermost oceanic crust covered by km-thick trench metasediments[26]. We selected 10 pristine samples from different sequences within a subducted oceanic slab (2 metapelites, 5 metabasites, 3 serpentinites, Supplementary Table 1) to obtain a general picture of sulfur reservoirs. Mineral assemblages suggest that one metapelite reached low blueschist-facies conditions (300–400 °C, 1.0–1.5 GPa), whereas the other metapelite (garnet–glaucophane-bearing) reached blueschist-facies conditions (400–500 °C, 1.5–2.0 GPa)[27]. Metabasites with oceanic affinity are eclogites and blueschists (lawsonite relic identified) with peak metamorphic conditions clustering around 540 °C and 2.5 ± 0.2 GPa[26]. Sulfides in all HP metapelites and metabasites are mainly pyrite with minor amounts of chalcopyrite and bornite[17]. Sulfide occurs both as inclusions in garnet and in the matrix. Matrix pyrite contains garnet, omphacite, glaucophane, lawsonite and dolomite inclusions (Supplementary Table 1). Serpentinites, composed mostly of antigorite and magnetite with minor pentlandite and millerite, are considered to be part of the subducted slab that underwent UHP metamorphism (~520 °C, >3.0 GPa)[28] in the Tianshan.

In addition, three representative sulfide-bearing dehydration-related veins in blueschists or eclogites were investigated to reconstruct the sulfur behavior in subduction fluids derived from different sources (Fig. 1). Vein_1 (JTS) consists of a well-studied wallrock–selvage–vein system[25,29] formed by fluid–rock interaction during prograde metamorphism (Fig. 1a). The wallrock (host blueschist, garnet–glaucophane-dominated) along the vein traverse was progressively altered to an eclogite selvage (garnet–omphacite-

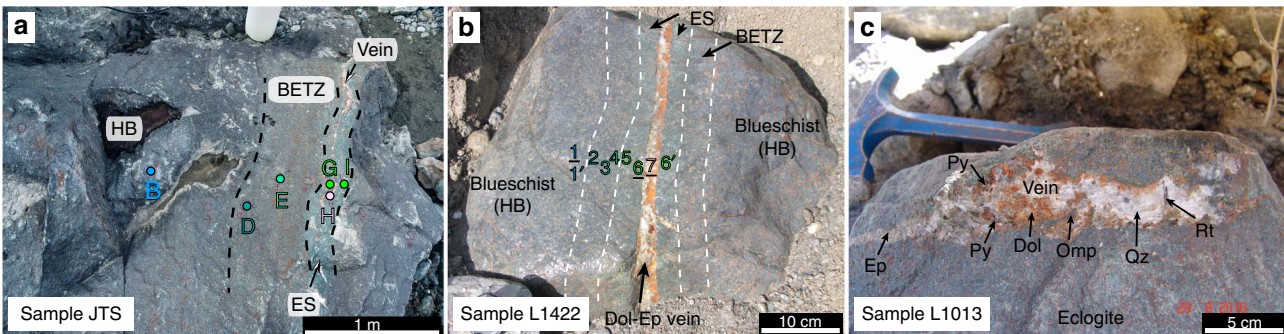

**Fig. 1 Field photographs of three sulfide-bearing veins in blueschists/eclogites (SW Tianshan). a** Sample JTS containing host blueschist (HB), blueschist–eclogite transition zone (BETZ), eclogite selvage (ES) and vein. Six drill samples (JTS-B, -D, -E, -G, -H, -I) along the traverse were taken for petrological investigation and chemical measurements. **b** HB–BETZ–ES–vein system L1422. Nine drill cores (see numbers) along the profile were investigated in this study. Petrological features of underlined drill samples are shown in Fig. 2. **c** Eclogite–vein sample L1013. Mineral abbreviations: dolomite (Dol), epidote (Ep), omphacite (Omp), pyrite (Py), quartz (Qz), and rutile (Rt).

dominated) and a blueschist–eclogite transition zone due to reaction with an external fluid (Fig. 1a). The wallrock–selvage–vein system equilibrated at peak metamorphic conditions of ~510 °C and 2.1 GPa[29]. Sr and Ca isotope compositions trace the fluid source to seawater-altered lithospheric slab–mantle and/or oceanic crust[25]. Vein_2 (L1422) is a 2-cm-wide dolomite–quartz–epidote-dominated vein crosscutting a massive host blueschist (Fig. 1b). Similar to Vein_1, the blueschist–eclogite transition zone and eclogite selvage formed due to interaction with Ca-rich fluid along the conduit (Fig. 1b). The similar structure, mineral assemblages, and compositions of Vein_1 and Vein_2 indicate that they formed at similar P–T conditions. Vein_3 (L1013) is a 1–3 cm wide

dolomite–quartz–epidote-dominated vein cutting massive eclogite (Fig. 1c). Occurrences of high-pressure minerals such as omphacite and rutile in Vein_3 along with the observation that no reaction halo occurs between the vein and host eclogite (Fig. 1c) indicate that the vein also formed at eclogite-facies conditions. Considering most eclogite samples in the Tianshan HP metamorphic belt were exhumed from ~80 km depth[26], all three HP veins are thought to represent the fluid activity that took place at 70–90 km depths in the subduction zone.

Sulfides are found in all vein samples. In sample JTS, pyrite is the dominant sulfide inclusion in garnet, but pyrrhotite dominates in the matrix (Fig. 2a, b). This demonstrates a pyrite–pyrrhotite transition

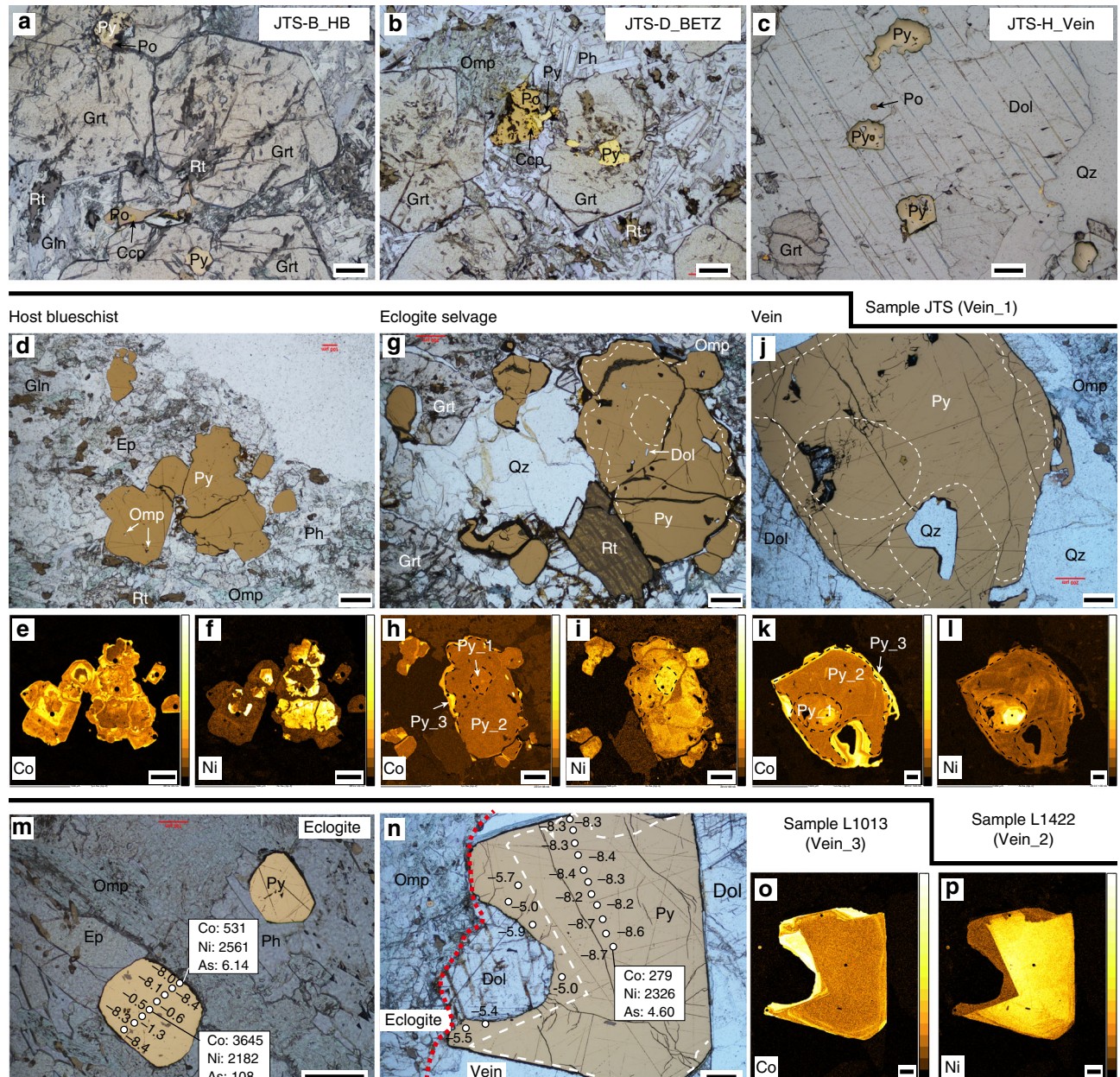

**Fig. 2 Photomicrographs and element mapping of sulfides. a–c** Sulfides in sample JTS. Sulfides in garnet are mainly pyrite with minor pyrrhotite and chalcopyrite, whereas the matrix contains mainly pyrrhotite with minor chalcopyrite and pyrite in both the host blueschist (**a**) and blueschist–eclogite transition zone (**b**). **d–i** Photomicrographs and Co–Ni maps of pyrite in the host blueschist (**d–f**), eclogite selvage (**g–i**), and vein (**j–l**) in sample L1422. **m–p** Pyrite, δ[34]S values, and Co–Ni maps in sample L1013. Trace-element concentrations (μg g[−1]) of pyrite are given in the white rectangles. δ[34]S values and Co–Ni maps in vein pyrite (**n–p**) suggest two growth generations. All photomicrographs (except Co–Ni maps) use superposed transmitted and reflected light to simultaneously image silicates and sulfides. Mineral abbreviations: chalcopyrite (Ccp), dolomite (Dol), epidote (Ep), garnet (Grt), glaucophane (Gln), omphacite (Omp), phengite (Ph), pyrite (Py), pyrrhotite (Po), quartz (Qz), and rutile (Rt). Scale bar: 200 μm.

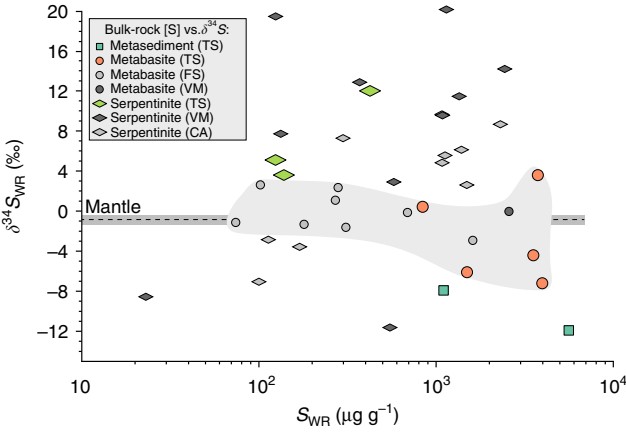

**Fig. 3 Whole-rock (WR) sulfur concentrations and isotope values of HP metasediments, blueschists/eclogites, and serpentinites worldwide.**
Errors (2σ) of $\delta^{34}S_{WR}$ and $S_{WR}$ calculated from measurement reproducibility are less than the symbol size. Sulfur isotope values of HP Tianshan (TS) rocks are from this study, Franciscan (FS) eclogites and blueschists from ref. [10], high-pressure serpentinites of the Voltri Massif (VM, Italy) from refs. [47,48], high-pressure serpentinites of the Cerro del Almirez (CA, Spain) from ref. [21], and the mantle value (−0.91 ± 0.50‰) from ref. [45]. The shaded area indicates $\delta^{34}S$ range of metabasites, which is comparable to the mantle value. Source data are provided in Supplementary Data.

during garnet growth due to changing sulfur fugacity–oxygen fugacity ($fS_2$–$fO_2$) conditions associated with fluid metasomatism. Sulfide in the vein is mostly pyrite but also includes minor pyrrhotite (Fig. 2c). Along the traverse of samples L1422, sulfide (mainly pyrite) abundances increase toward the vein. The Co–Ni element distribution maps (Method 1) reflect distinct differences between host blueschist pyrite (Fig. 2d–f) and selvage–vein pyrites (Fig. 2g–i). Both selvage and vein pyrites display multiple growth generations (Fig. 2g–i). In sample L1013, Co–Ni element distribution maps and contents show core–rim textures in both eclogite (Fig. 2m) and vein pyrite (Fig. 2n–p).

In most samples, fine fractures were occasionally observed surrounding sulfide grains, reflecting rigidity contrasts between sulfide and matrix minerals. These fractures are usually filled with albite + magnetite + calcite ± chalcopyrite ± barite due to late-stage fluid infiltration, accompanying variable retrogression of neighboring omphacite and glaucophane. Sulfate and magnetite were observed only in these late-stage, retrograde fractures.

**Bulk-rock sulfur geochemistry of different lithologies.** The 10 HP rocks and 15 representative subsamples from two host–selvage–vein systems (JTS and L1422) were analyzed for their whole-rock sulfur contents ([S]$_{WR}$) and $\delta^{34}S$ compositions ($\delta^{34}S_{WR}$, Method 1). The metapelites have [S]$_{WR}$ = 1101–5612 µg g$^{-1}$ and negative $\delta^{34}S_{WR}$ of −12 to −7.9‰ (Fig. 3, Supplementary Table 1) and the metabasites have [S]$_{WR}$ = 841–3978 µg g$^{-1}$ and a range in $\delta^{34}S_{WR}$ of −7.2 to +3.6‰, averaging −2.7‰ ($n = 5$). In contrast, all measured serpentinites ([S]$_{WR}$ = 124–422 µg g$^{-1}$) have positive $\delta^{34}S_{WR}$ values (+3.6 to +12‰), signifying high-temperature water–rock interaction during oceanic serpentinization[30]. These results are similar to unmetamorphosed oceanic lithosphere[19–21,30] and suggest that the main slab sulfur reservoirs have distinct $\delta^{34}S_{WR}$ compositions, which are generally consistent with previous studies of exhumed slab rocks[14,15] (Fig. 3). Sulfur in nearly all samples is present as reduced S$^{2-}$ or S$^-$, whereas S$^{6+}$ contents are very low ([S$^{6+}$]/[S]$_{WR}$ < 0.06). Only one serpentinite sample has more sulfate than sulfide ([S$^{6+}$]/[S]$_{WR}$ ≈ 0.89).

**Sulfur geochemistry in HP vein systems.** In the host blueschist to blueschist–eclogite transition zone of Vein_1 the [S]$_{WR}$ varies between 698 µg g$^{-1}$ and 841 µg g$^{-1}$, but toward the vein and vein-like eclogite selvage, the [S]$_{WR}$ increases up to 2183 µg g$^{-1}$ (Fig. 4a). In contrast, the $\delta^{34}S_{WR}$ values decrease gradually from +0.43‰ to −0.98‰ toward the vein (Fig. 4a). The in situ $\delta^{34}S$ values of sulfides in garnet show little variation; however, matrix sulfides display a decreasing trend from host rock toward the vein (Fig. 4b; Method 1). The local bulk isotopic compositions of sulfides ($\delta^{34}S_{sulfide}$), calculated using mean in situ $\delta^{34}S$ values of individual sulfides (Fig. 4b) and their mineral volume ratios (Supplementary Fig. 1), have a narrow range of +0.34–+0.72‰ (mean +0.60‰) in the garnet along the traverse (Fig. 4b). This reflects shielding of the sulfide inclusions by garnet during fluid–rock interaction. In contrast, the $\delta^{34}S_{sulfide}$ in the matrix decreases gradually from about 0.00‰ to −1.35‰ toward the vein (Fig. 4b). The $\delta^{34}S_{WR}$ and $\delta^{34}S_{sulfide}$ display similar decreasing trends along the traverse. Vein sulfides have uniform $\delta^{34}S_{sulfide}$ values of about −1.0‰ (Fig. 4b).

The host blueschist of Vein_2 has the lowest [S]$_{WR}$ of 604 µg g$^{-1}$ and $\delta^{34}S_{WR}$ values of −11.0‰ (Fig. 5a, Supplementary Table 1). Within the blueschist–eclogite transition zone, both [S]$_{WR}$ and $\delta^{34}S_{WR}$ increase and have relatively narrow ranges of 1465–1854 µg g$^{-1}$ and −8.1 to −7.7‰, respectively. These values increase further to [S]$_{WR}$ = 2167–2848 µg g$^{-1}$ and $\delta^{34}S_{WR}$ = −1.7 to −0.9‰ in the eclogite selvage (Fig. 5a). The vein has the highest [S]$_{WR}$ (9251 µg g$^{-1}$) and $\delta^{34}S_{WR}$ value (−0.7‰) (Fig. 5a). In situ pyrite $\delta^{34}S$ compositions (Fig. 5b–d) are consistent with the bulk-rock analyses. Pyrite in the host blueschist has Ni-rich cores with negative $\delta^{34}S$ values of −16.0 to −10.1‰ (weighted mean −12‰) and Ni-poor rims with $\delta^{34}S$ values of −7.7 to −5.0‰ (weighted mean −7‰) (Fig. 5b). The vein and selvage pyrites show uniform core–mantle–rim textures recorded by Co–Ni element distribution maps (Fig. 2g–l): a Ni-rich core with modest enrichment in Co (Py_1, Co: 367 µg g$^{-1}$, Ni: 1277 µg g$^{-1}$, $\delta^{34}S$: +7.4 to +8.6‰); a massive Co-poor and moderately Ni-enriched mantle (Py_2, Co: 174 µg g$^{-1}$, Ni: 813 µg g$^{-1}$, $\delta^{34}S$: +0.9 to +4.0‰); and a thin Co-rich and Ni-poor rim (Py_3, Co: 3803 µg g$^{-1}$, Ni: 229 µg g$^{-1}$, $\delta^{34}S$: −6.7 to −2.3‰) (Fig. 5c, d). These three pyrite generations with variable Co–Ni contents and $\delta^{34}S$ values likely represent three stages of fluid infiltration.

The vein pyrite of Vein_3 has a thick Co-poor but Ni-rich core ($\delta^{34}S$ = −8‰) and a thin Co-rich but Ni-poor rim ($\delta^{34}S$ = −5‰) (Figs. 2n–p, 5f). In contrast, pyrite in the host eclogite contains a Co–Ni-rich core with MORB-like $\delta^{34}S$ values (−1.3 to −0.5‰), but its rim is analogous to the vein pyrite core in Co–Ni–As contents, $\delta^{34}S$ values (about −8‰), and mineral inclusions (omphacite and rutile) (Figs. 2m, 5e). This indicates that the vein-forming fluid also altered the immediate eclogite and caused pyrite regrowth surrounding the cores.

**Sulfur concentrations in aqueous fluids from DEW modeling.** The sulfur concentration in fluids ([S]$_{fluid}$) is the most important factor determining the slab sulfur output, as aqueous fluids are thought to be the major agent for slab–mantle sulfur transfer[11]. We use the DEW (Deep Earth Water) model[31,32] to calculate subduction zone [S]$_{fluid}$ (Method 2), as this allows a quantitative prediction of speciation and solubilities in fluids at upper mantle conditions. Because sulfur solubility and speciation is redox-dependent, an estimate of the $fO_2$ is required prior to calculation. The $fO_2$ of subducted AOC is FMQ + 1 (ref. [33]) (one log unit above Fayalite–Magnetite–Quartz buffer) at the trench and decreases gradually with increasing depth (below FMQ at depths corresponding to eclogite-facies conditions)[17], as generally reducing fluids (<FMQ) are generated[17,34]. In contrast, the redox

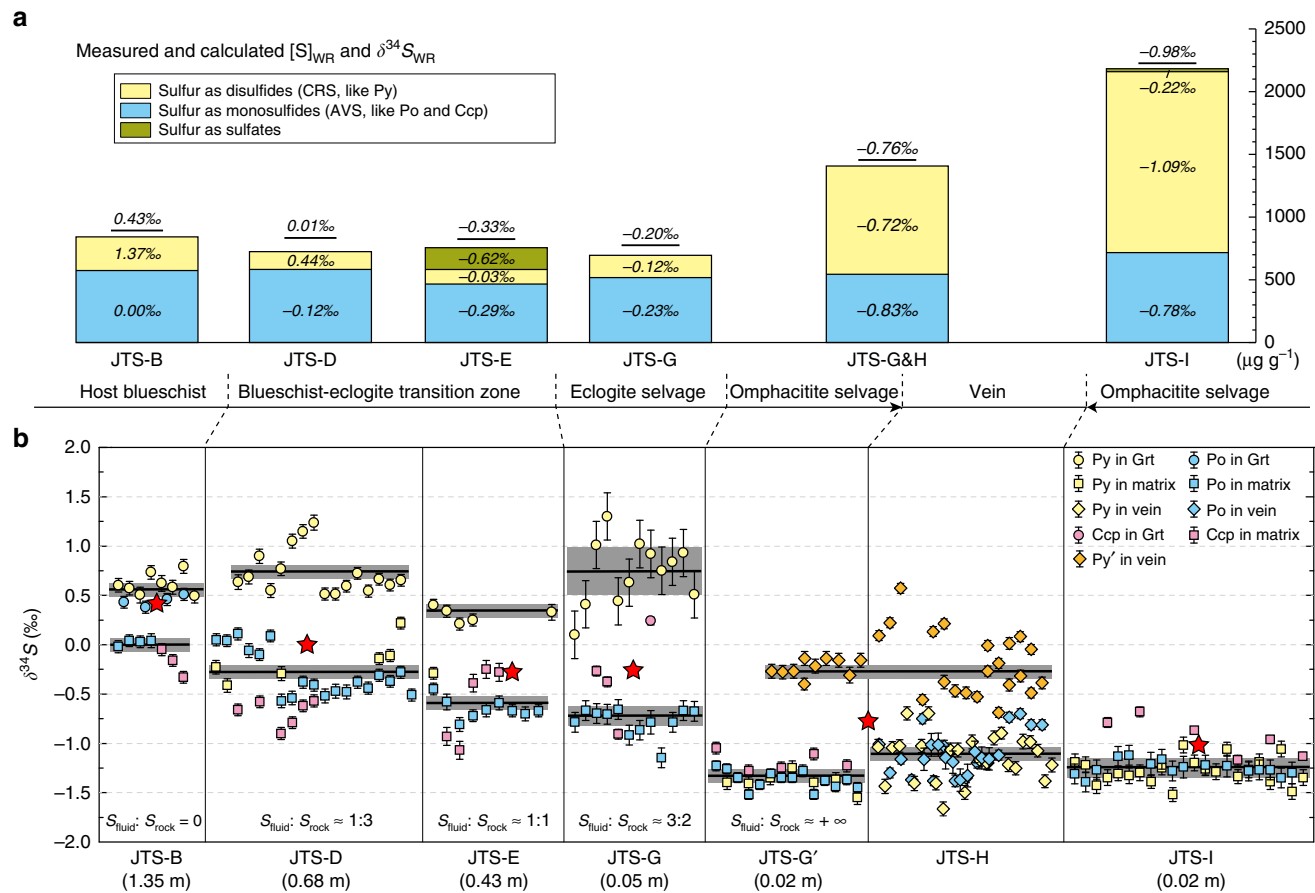

**Fig. 4 Variations of bulk-rock and in situ sulfide $\delta^{34}S$ compositions along the traverse of sample JTS (Vein_1). a** Calculated $[S]_{WR}$ and $\delta^{34}S_{WR}$ compositions based on measured whole-rock (WR) compositions of acid volatile sulfide (AVS), chromium reducible sulfide (CRS) and sulfate (Method 1). Rectangle height indicates sulfur content, and underlined $\delta^{34}S$ values above rectangles refer to calculated $\delta^{34}S_{WR}$ compositions. Sulfate contents are mostly below the detection limits or very low. **b** In situ $\delta^{34}S$ compositions of sulfides along the profile. Sulfides occurring as inclusion in garnet are denoted by circles, matrix sulfides by squares, and vein sulfides by diamonds. Bold black lines refer to calculated $\delta^{34}S_{sulfide}$ values (see also Supplementary Fig. 1). Vein sulfides include two types of pyrite. Texture investigation indicates that one type (orange diamonds) is late-stage with $\delta^{34}S$ around $-0.21‰$, which was not included in the calculation for JTS-G' and JTS-H. The vertical bars show the analytical errors calculated after propagating the within-run and external uncertainties of standard measurements. Red stars show $\delta^{34}S_{WR}$ compositions from **a**. Mineral abbreviations: chalcopyrite (Ccp), garnet (Grt), pyrite (Py), and pyrrhotite (Po). Source data are provided in Supplementary Data.

state of slab serpentinite is more complicated (either above or below FMQ) and is suggested to vary due to different degrees of pre-subduction serpentinization[35], producing both highly oxidizing or reducing fluids[35]. Dehydration of incompletely serpentinized rocks (usually those beneath oceanic crust) in which awaruite is present produces reducing $H_2$-bearing fluids, whereas deserpentinization of completely serpentinized rocks (usually those once directly exposed to seawater) in which awaruite is absent produces oxidizing fluids in the subduction zone[35]. The former is applicable in our case, as the majority of slab mantle occurs beneath oceanic crust and is not fully serpentinized. For details regarding slab $fO_2$ estimates see the Supplementary Note 1.

Following a typical subduction geothermal gradient[36], [S] and its speciation in slab fluids were calculated for given $fO_2$ conditions at 60 km (FMQ), 75 km (FMQ), 90 km (FMQ-1), 120 km (FMQ-2), and 150 km (FMQ-3) for subducted sediments and oceanic crust, whereas $fO_2$ was 1–2 log units higher for serpentinites at the corresponding depths (Supplementary Table 2). Results show that $[S]_{fluid}$ is largely dependent on P–T conditions (Fig. 6a) and is generally very low (<0.1 molal), similar to previous thermodynamic modelling[37–39]. Critically, however, our results reveal a distinct $[S]_{fluid}$ peak (0.20–0.35 molal) at ~3.0 GPa, regardless of whether the fluids equilibrated with

metasediments, metabasalts or serpentinites (Fig. 6a), indicating a sulfur release pulse at ~90 km depth. Sulfur species are $fO_2$-dependent and dominated by reduced aqueous $H_2S$ and $HS^-$ at all model subduction zone P–T–$fO_2$ conditions (Supplementary Fig. 2), consistent with our natural observations. Oxygen fugacity variations of ±1 unit will only change the proportion of sulfur species in the fluid (e.g., slight increases of $SO_4^{2-}$ and/or $HSO_4^-$ abundances), but will not cause significant $[S]_{fluid}$ changes (Supplementary Fig. 2).

**Hydrothermal sulfur isotope fractionation.** Effects of sulfur isotope fractionation during hydrothermal processes are largely influenced by pressure, temperature, $fO_2$ and the pH of the fluids[40,41]. We thermodynamically calculated $fO_2$–pH diagrams[42] (Fig. 7a–d) (Method 3) for different P–T conditions to reveal $\delta^{34}S$ fractionation in subduction zones. Our DEW calculations suggest that slab fluids are generally alkaline and the pH value ranges from neutral ($pH_n$) to $pH_n + 2$, consistent with previous work[43]. According to $fO_2$ (Supplementary Table 2) and pH estimates of subduction zone fluids, all the $fO_2$–pH conditions plot in fields dominated by the species $H_2S$ or $HS^-$ (yellow area, Fig. 7a–e), in agreement with DEW results (Supplementary Fig. 2). The $fO_2$–pH diagram indicates limited sulfur isotope fractionation

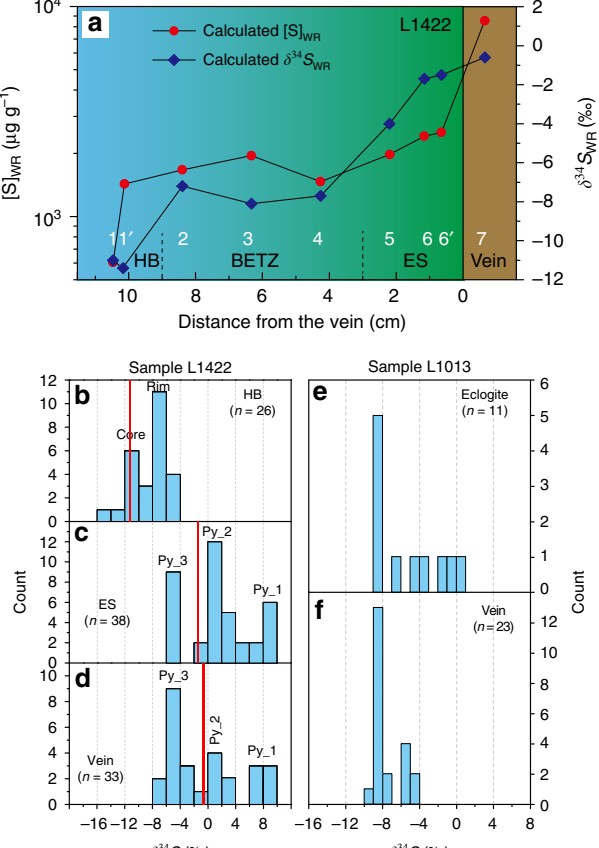

**Fig. 5 Whole-rock (WR) and in situ pyrite δ34S compositions of vein samples. a** Calculated $[S]_{WR}$ and $\delta^{34}S_{WR}$ compositions from bulk-rock analyses along the traverse of sample L1422. Errors of $\delta^{34}S_{WR}$ and $S_{WR}$ calculated from analytical reproducibility are smaller than the symbols used. HB host blueschist, BETZ blueschist–eclogite transition zone, ES eclogite selvage. **b** Histograms of in situ pyrite δ34S values in the HB, ES, and vein (sample L1422, red lines refer to the $\delta^{34}S_{WR}$ values). **c** Histograms of pyrite δ34S values in the sample L1013. Source data are provided in Supplementary Data.

(<3‰) at different P–T conditions along the subduction interface (Fig. 7a–d). In particular, at the vein-forming P–T conditions of this study, $fO_2$ (<FMQ)[17] and pH range ($pH_n$ to $pH_n + 2$) suggest sulfur isotope fractionations <1.3‰ (Fig. 7e). In addition, precipitation styles of sulfide in hydrothermal settings (closed- or open-system) may also influence the sulfur isotope fractionation[41]. In both closed and open systems, theoretical calculations (Method 3) display <1‰ fractionation if pyrite precipitated from $H_2S^-$-dominated fluids at 550 °C (Fig. 7f, g), consistent with previous calculations for subduction conditions[14,15].

**δ34S values of fluids from different slab reservoirs.** The small sulfur isotope fractionation between sulfides and equilibrated $H_2S$-bearing fluids[40,41] (Fig. 7) demonstrates that the δ34S values of vein sulfides approximately represent the fluid δ34S composition and can be used as a tracer for source discrimination. Our measured negative $\delta^{34}S_{WR}$ values of −12 to −8‰ in metasediments (Fig. 3) are similar to their protoliths, the young marine sedimentary rocks (Phanerozoic) that mostly have δ34S values of −24 to −8‰ due to the presence of biogenically produced sulfide[19,44]. It is suggested that sulfide in sediments retains its δ34S characteristics during subduction metamorphism[13] and that metasediments may act as a negative δ34S reservoir in subducting

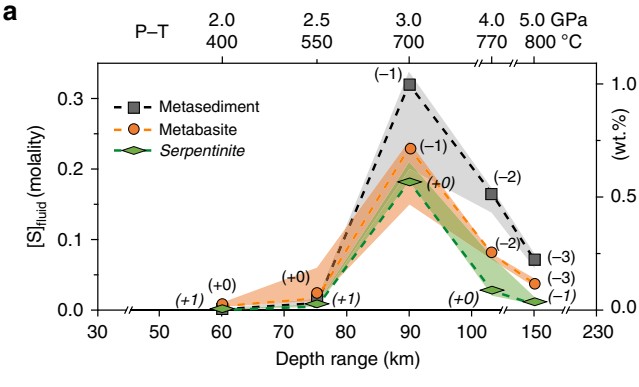

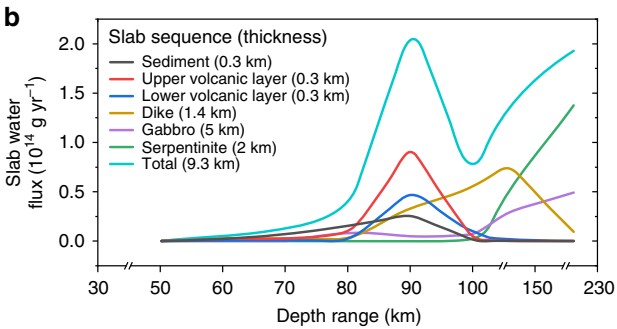

**Fig. 6 Sulfur concentrations in fluids and slab water fluxes used for subduction sulfur output estimate. a** $[S]_{fluid}$ derived from different lithologies in the subduction zone calculated by DEW model. Numbers in brackets refer to oxygen fugacity (relative to FMQ buffer) used at certain P–T conditions. Shaded areas indicate $[S]_{fluid}$ variations if $fO_2$ changes within ±1 unit. Sulfur species and proportions at every point are given in Supplementary Fig. 2. **b** Depth-dependent water flux released from global subducting slabs[49]. Water flux is calculated every 10 km between 50–100 km. For example, about $0.91 \times 10^{14}$ g yr$^{-1}$ $H_2O$ is dehydrated from the upper volcanic layer over the depth interval 80–90 km (corresponding to X-axis 90 km). Source data are provided in Supplementary Data.

slabs. The vein pyrite core with negative δ34S (−8‰) in sample L1013 (Fig. 2n) thus likely represents the δ34S signature of fluids derived from the abundant subducted metasediments in the Tianshan HP belt[16,26].

Pristine oceanic crust is typically within the range of the average mantle δ34S of −0.91 ± 0.50‰[45]. However, pre-subduction seafloor processes lead to considerable δ34S heterogeneities in the upper crust (Fig. 3, Supplementary Table 1). For instance, microbial sulfate reduction during seafloor alteration moves the δ34S of the volcanic section toward negative values (average −6‰), some as low as −19.5 to −45‰[22]. The blueschists studied herein with more negative $\delta^{34}S_{WR}$ (−15‰ and −11‰, Supplementary Table 1) may record this microbially produced δ34S heterogeneity. In contrast, sulfide grains in some eclogites from the Alps and New Caledonia show positive δ34S compositions (+7‰ and +12‰)[15]. However, these sulfides are associated with blueschist/greenschist retrogression[15], which may record the oxidizing fluids at shallow depth that usually cause retrogression of exhumed eclogites/blueschists[17]. The positive δ34S may originally come from seawater hydrothermal alteration, or represent the fluids derived from serpentinite dehydration (see below). Therefore, although pre-subduction seafloor processes cause negative or positive δ34S shifts in metabasites, bulk-rock geochemistry shows that many mafic eclogites and blueschists still retain their mantle-like sulfur isotope signature throughout HP metamorphism (Fig. 3). Furthermore, the lower crust (dike and gabbro) retains its mantle-like δ34S values as well[46].

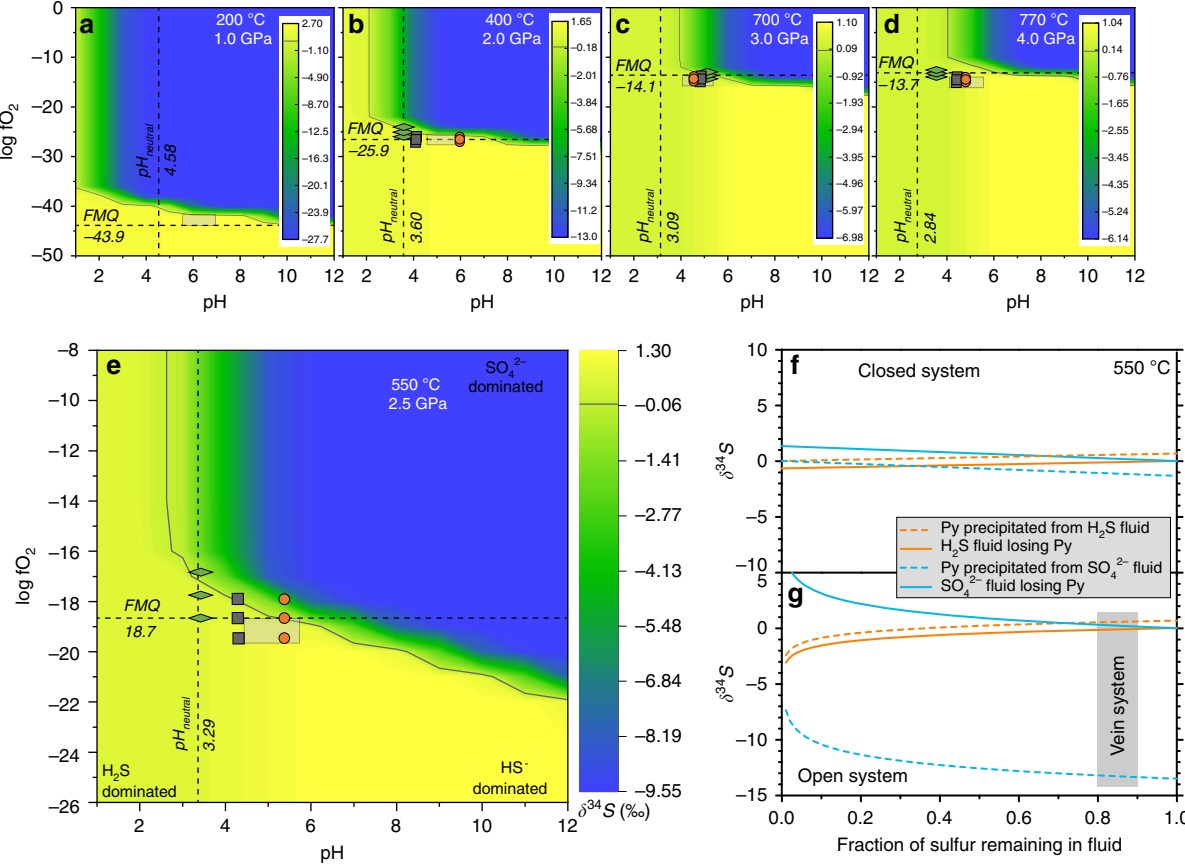

**Fig. 7 Thermodynamic modelling of sulfur isotope fractionation as a function of temperature, $fO_2$ and pH. a–d** Influence of $fO_2$ and pH on the isotopic compositions of pyrite precipitated from hydrothermal fluids at ionic strength $I = 1.0$ and $\delta^{34}S_{\sum S} = 0‰$, at specific P–T conditions of various depths along the subduction interface. The squares (metasediment), circles (metabasalt) and diamonds (serpentinite) represent estimated $fO_2$ conditions (Supplementary Note 1) and calculated pH values from the DEW model. The rectangles denote slab $fO_2$–pH conditions from ref. [43]. The color scales (‰) refer to $\delta^{34}S$ contours as a function of $fO_2$ and pH, and the gray solid line indicates the contour with no sulfur isotope fractionation. **e**, $\delta^{34}S$ fractionation modelling at P–T conditions of the vein samples studied here. **f–g** Sulfur fractionation calculation plots of $FeS_2$ precipitation from fluids at 550 °C in a closed system (**f**) and an open system following a Rayleigh fractionation model (**g**). Solid lines denote the fluid phase, whereas the dashed lines represent the precipitated phase. The orange and blue lines represent the results for $H_2S$- and $SO_4^{2-}$-fluids, respectively. Initial $\delta^{34}S_{fluid}$ is 0‰. The shaded area indicates the expected sulfur fraction of sulfide precipitation under a channelized fluid flux. Source data are provided in Supplementary Data.

Consequently, the $\delta^{34}S$ of vein sulfides from sample JTS ($-1‰$) is interpreted to record the sulfur isotope signature of fluids released from the oceanic crust (the dike and gabbro part, in particular). This fluid source interpretation is consistent with the $\delta^{34}S_{WR}$ of the host blueschist ($+0.43‰$, Supplementary Table 1) and trace-element contents[29] and Sr–Ca isotope compositions[25] of the vein.

Serpentinites are quite heterogeneous in [S], $S^{6+}/\Sigma S$, and bulk-rock $\delta^{34}S$ (refs [21,30]), and are readily influenced by late-stage fluids during exhumation[23]. Our three measured Tianshan UHP serpentinites all have positive $\delta^{34}S_{WR}$ compositions, consistent with bulk-rock results for Voltri Massif serpentinites[21,47,48] (Fig. 3) and in situ sulfide $\delta^{34}S$ compositions from Corsican serpentinites[14]. We suggest that variably serpentinized slab mantle beneath oceanic crust is characterized by positive $\delta^{34}S$ compositions as a result of sulfide addition via sulfate reduction at high-temperatures[30] during partial serpentinization. Thus, the Py_1 with positive $\delta^{34}S$ ($+8‰$) in vein sample L1422 (Figs. 2k, 5d) is interpreted to reflect the characteristic $\delta^{34}S$ composition of fluids derived from the partly serpentinized slab mantle. The pyrite mantle (Py_2) with positive but decreasing $\delta^{34}S$ (from $+4.0$ to $+0.9‰$) of vein sample L1422 (Figs. 2k, 5d) likely represents fluid mixing with an increased AOC contribution relative to slab serpentinites. The negative $\delta^{34}S$ ($-5‰$) of thin

rims on vein pyrite (Figs. 2n, 5d) with the sharply increasing Co concentrations (Fig. 2k, o) may represent retrograde oxidized fluids during exhumation[17], as evidenced by the surrounding fractures with albite-calcite-magnetite infillings and neighboring retrogression of matrix omphacite.

## Discussion

Thermodynamic modelling shows that at subduction zone P–T–$fO_2$–pH conditions sulfur in fluids is dominated by the reduced $H_2S$ and $HS^-$ species, whereas sulfate species (e.g. $SO_4^{2-}$, $HSO_3^-$) are rare (Supplementary Fig. 2). This is consistent with our petrological evidence for the occurrence of sulfide, but not sulfate, in the veins (Figs. 1–2), the very low sulfate concentrations in rocks and veins (Fig. 4a; Supplementary Table 1), and previous experimental results[11]. If slab fluids are dominated by sulfate as some recent studies propose[13], several predictions follow. First, the oxidizing fluid will produce a redox gradient in the immediate wallrock, but this is not recorded in the selvages we examined. Second, reduction from $S^{6+}$ to $S^{2-}$ will cause oxidation within the immediate rock and vein (in particular during vein crystallization) in the form of hematite or magnetite[17], which, however, are absent from the veins or selvages. Third, complete sulfate–sulfide transformation will produce very high $\delta^{34}S$ values in the product phases, which is also not observed in the veins or

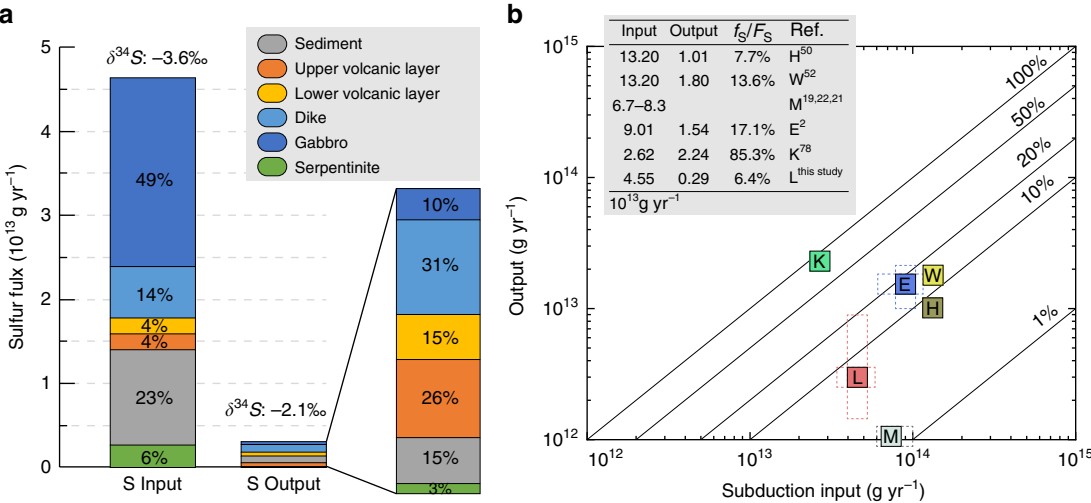

**Fig. 8 Mass-balance calculation results of subduction sulfur input and output. a** Subduction sulfur input and output (30–230 km) categorized by source. **b** Comparison of subduction sulfur input *vs.* output. Sulfur influx and outflux estimations are given in the inset table. Labeling of estimates refers to source data (see inset), E to Evans[2]; H to Hilton et al.[50]; K to Kagoshima et al.[78]; L to this study; W to Wallace[51]; and M to Canfield[19], Alt and Shanks[22], and Alt et al.[21]. Uncertainties of estimates indicated with dashed rectangles. No errors are shown if they were not provided in the original study. Note that there is no output data for M.

selvages. The sulfate introduced during pre-subduction hydrothermal seafloor alteration[20,21,30] may have been lost or converted to sulfide at early stages of subduction, for example, at fore-arc depths[10,12]. Thus, we conclude that the dehydration-related slab fluids likely transport reduced sulfur species such as aqueous $H_2S$ and $HS^-$ at sub-arc depths.

Mass-balance calculations were used to estimate the sulfur influx ($F_S$) and outflux ($f_S$) of subduction zones, as well as their $\delta^{34}S$ values. The sulfur input estimate was computed (Method 4) based on the average [S] and $\delta^{34}S$ compositions of our best current understanding of oceanic lithosphere stratigraphy (Supplementary Fig. 3, Note 2) in combination with the global length of subduction zones and their average convergence rate, sequence thickness, and density. The resulting subduction zone sulfur influx is estimated to be $4.65 \times 10^{13}$ gram per year (g yr$^{-1}$) with a bulk negative $\delta^{34}S$ value of −3.60‰. Gabbro (49%) and sediment (23%) are two important sulfur reservoirs (Fig. 8a), whereas serpentinite is insignificant due to its low [S]. This influx is slightly lower than, but generally within the same order of magnitude of, previous estimates[2,19,21,22] (Fig. 8b).

The sulfur output can be calculated from the product of [S]$_{fluid}$ and the fluid fluxes from the dehydrating slab. Previous thermodynamic constraints[37–39] on [S]$_{fluid}$ predict a rather low proportion of $H_2S$ (<0.01 mol.% in equilibrium with $H_2O$ + pyrite + pyrrhotite)[38,39]. This low [S]$_{fluid}$, in turn, predicts negligible slab sulfur output ($f_S/F_S$ < 1%), which is not compatible with the high sulfur contents observed in arc settings. Based on our novel [S]$_{fluid}$ results from the DEW calculations (Fig. 6a) and depth-dependent fluid fluxes from the dehydrating slab[49] (Fig. 6b), a total sulfur outflux (Method 5) of $2.91 \times 10^{12}$ g yr$^{-1}$ (6.3% of $F_S$) is estimated via fluids derived from the subducting slab between 30–230 km (Fig. 8a). The sheeted dikes (31%) and the upper volcanic layer (26%) contribute most of the sulfur release, but the sediment (15%) and the lower volcanic layer (15%) are also important (Fig. 8a). This sulfur outflux from the slab is about 1/5 to 1/3 of the sulfur output estimates from arcs[2,50,51] (Fig. 8b).

Importantly, our sulfur output estimate shows a major sulfur release of $2.46 \times 10^{12}$ g yr$^{-1}$ (5.3% of $F_S$) to the mantle wedge at depths of 70–100 km (Fig. 9) due to both elevated [S]$_{fluid}$ (Fig. 6a) and released $H_2O$ flux[49] (Fig. 6b). The volcanic layers, dikes, gabbro, and sediments contribute to the major sulfur release at

this depth interval (Fig. 8a). This sulfur release window coincides with pyrite-to-pyrrhotite breakdown[12] (releasing $H_2S$) and the major slab fluid release (~32% $H_2O/H_2O_{total}$)[49], which subsequently acts as the trigger for partial melting of the mantle wedge and ultimately arc magmatism[25,52,53]. The release of slab sulfur is minor at other depths (Fig. 9). The DEW model may underestimate [S]$_{fluid}$ due to unknown sulfur species. In addition, considering the uncertainties on the thermodynamic data (±0.3 to ±0.5 of logK)[31], the slab sulfur loss estimate may extend to a maximum of ~20%. This is comparable to the estimate obtained from natural rocks and experiments (30%, Supplementary Note 3), and previous estimates (8–18%, Fig. 8b) from arcs[2,50,51].

Knowing [S]$_{fluid}$ (Fig. 6a) and water fluxes[49] (Fig. 6b) liberated from slab sequences as well as the $\delta^{34}S_{fluids}$ of sediments (−8‰), oceanic crust (−1‰) and serpentinites (+8‰), mass-balance calculations (Method 5) predict a $\delta^{34}S$ value of slab fluids of −1.8 ± 3‰ at depths of 70–100 km. Both the gradually decreasing $\delta^{34}S_{WR}$ and $\delta^{34}S_{sulfide}$ values (sample JTS, Fig. 4a) and the trend of increasing $\delta^{34}S_{WR}$ values within the studied profile (sample L1422, Fig. 5a) provide clear evidence for isotope exchange during fluid–rock interaction, despite rapid fluid transport rates[24,25]. The sulfur isotope composition of channelized fluids generated in the lower parts of a slab could be slightly altered within the upper parts during fluid transport. Assuming a 10% sulfur isotopic contamination rate by sediment and 20% by oceanic crust for underlying sulfur sources, the net $\delta^{34}S$ value is calibrated to −2.5 ± 3‰ at depths of 70–100 km (Fig. 9). If $\delta^{34}S_{fluids}$ values from different reservoirs are kept constant at different depths, $\delta^{34}S$ of slab fluids at shallower depths remains negative but shifts to positive at greater depths, and a net $\delta^{34}S$ of −2.1‰ is obtained for slab fluids between the whole 30–230 km (Fig. 9). Regardless of flux uncertainties, we emphasize that the negative $\delta^{34}S$ value of slab fluids is robust and insensitive to a range of different model scenarios (Method 6), including thick serpentinized slab-mantle scenarios[54], which may supply more fluids with positive $\delta^{34}S$ values.

This study provides the first comprehensive and quantitative view of the flux and isotope compositions of sulfur-bearing slab fluids, which likely mirror the slab sulfur contributions to arcs[25,55]. We show that dehydration-related fluids transfer modest amounts of sulfur (6.4% of total subducted sulfur, up to 20% maximum)

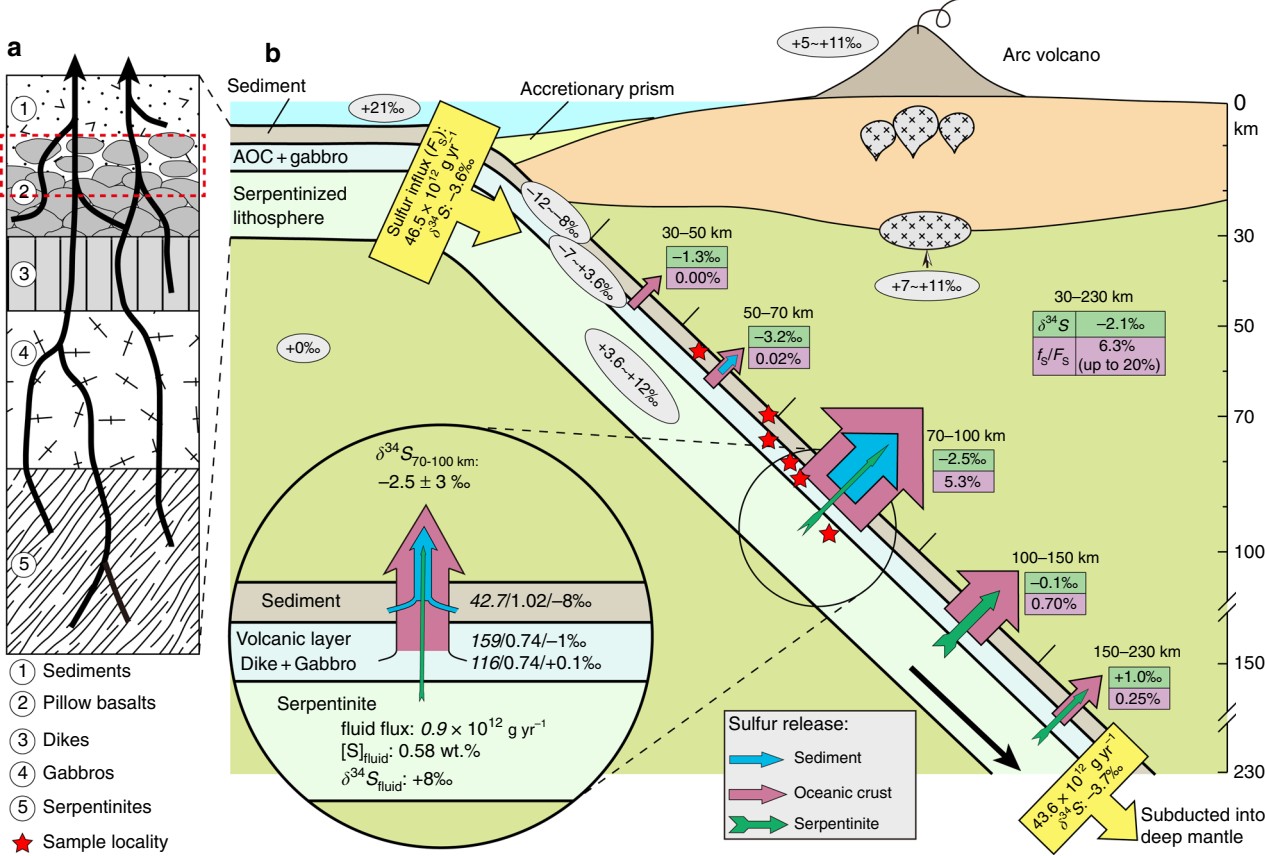

**Fig. 9 Diagram illustrating fluid-mediated sulfur release in the subduction zone. a** Schematic lithologic succession of typical subducted oceanic lithosphere. The bold arrows refer to channelized fluid flow, and the dashed rectangle refers to the sequences of this case study in the Tianshan. **b** Estimated sulfur flux (arrow sizes represent the relative sulfur amounts) and isotope compositions released from the subducting slab at different depths via fluid flow. Inset circle shows the key parameters during net $\delta^{34}S$ calculation of slab fluids to sub-arc mantle, such as global $H_2O$ flux, sulfur concentration and isotopic composition of fluids derived from different sequences of the subducting slab. $f_S/F_S$ ratio of 20% is the maximum value from DEW results. Numbers in ellipsoids refer to bulk sulfur isotope compositions in reservoirs of subduction settings (data sources see text). Red stars represent the depths of metasediments, metabasites (including veins) and serpentinites in this study formed in the subduction zone. Not to scale. Source data are provided in Supplementary Data.

from the slab to the mantle wedge. This maintains elevated sulfur contents of the mantle source for arc magmas (250–500 $\mu g\,g^{-1}$)[6,7] in comparison to MORB (80–300 $\mu g\,g^{-1}$)[56]. Additional significant release by, for example, slab melting is unlikely, as [S] in melts is much lower than in aqueous fluids ($D_S^{fluid/melt}$ usually >200)[11]. This slab-arc sulfur cycle is operated by fluid-mediated $H_2S$ and/or $HS^-$ transport with negative $\delta^{34}S$ composition, which has no direct links to the high oxygen fugacity and heavy $\delta^{34}S$ signature observed in arc volcanic rocks.

Our work also sheds further light on the nature of arc magmas. The reason for the higher $fO_2$ of arc magmas[57,58] relative to MORBs is still debatable. Intraoceanic or rare continental arcs, like those of the Western Pacific, may record flux melting; mantle peridotites with elevated $fO_2$ values in these settings have been thought to be influenced by slab-derived oxidizing agents[59,60]. In contrast, continental arcs, like those of the Eastern Pacific arcs where crust thickness may modulate the melting degree[61], may represent a complicated melting mode involving decompression and mantle peridotite that is not necessarily oxidized[59,62]. Direct comparisons of Western and Eastern Pacific arcs may be challenging due to their different melting modes and arc maturity[59,60]. The P–T evolution of the Tianshan eclogites (representing cold/old subducted oceanic slab) corresponds more closely to the thermal structure of subduction zones beneath the Western Pacific arcs[59]. However, our finding of negligible sulfate

in the slab fluids indicates that slab $SO_4^{2-}$ was unlikely to be the main oxidizing agent during South Tianshan Ocean subduction. In such environments, the high $fO_2$ of sub-arc mantle may instead result from addition of slab $H_2O$ and $CO_2$ (refs [63,64]), instead of oxidized sulfur species. Processes including incorporation of $H_2$ into orthopyroxene[63] and the formation of diamond[64] and CH4 (ref. [65]) in the mantle wedge may produce oxidized melts that elevate the $fO_2$ of Western Pacific arc magmas.

The calculated negative $\delta^{34}S$ ($-2.5 \pm 3‰$) released from the subducted slab (Fig. 9) contrasts with the positive $\delta^{34}S$ values found in the inclusions of Western Pacific arc rocks[4,5,8]. In general, the mantle wedge should have mantle-like $\delta^{34}S$ values of ~0‰. Therefore, the positive $\delta^{34}S$ signature in arc-related rocks requires additional sulfur sources or processes for $^{34}S$ enrichment. Volcanic degassing effects on melt $\delta^{34}S$ are highly dependent on redox state[66,67]. But even under oxidizing conditions (>FMQ + 2), increases in melt $\delta^{34}S$ caused by degassing are modest (~1.5‰)[66]. Therefore, the negative-to-positive shift in the $\delta^{34}S$ composition of melts must happen as part of the partial melting processes, such that significant sulfur isotope fractionation accompanies the melt oxidization. For example, $^{32}S$ may be scavenged into surrounding mantle to form sulfides while $H_2$ is incorporated into orthopyroxene[63], producing $^{34}S$-rich sulfate in oxidizing melt and finally isotopically heavier arc magmas. Thus,

further studies will be necessary to assess the processes that may lead to the positive $\delta^{34}S$ compositions in arc magmas.

Our comparison of subduction input with output fluxes indicates that most of the sulfur (>80%) with negative $\delta^{34}S$ values (<−3.7‰) is retained in the descending slab and recycled to the deep mantle (Fig. 9). This may have resulted in a progressive $^{34}S$-enrichment of Earth's surface sulfur reservoirs[19], and can explain the negative $\delta^{34}S$ values of alkaline magmas related to ocean island basalts (OIBs) since the Phanerozoic[68].

## Methods

**Analytical methods.** Bulk-rock sulfur contents and isotope compositions were measured at the Geological Institute at the Freie Universität Berlin. Extraction of the bulk-rock sulfur was performed by extracting the acid volatile sulfide (AVS), chromium reducible sulfide (CRS), and the sulfate fraction[69]. Sulfur isotope measurements of AVS, CRS, and sulfate fractions were done on a Thermo Fisher Scientific MAT 253 mass spectrometer combined with a Eurovector elemental analyzer. The $[S]_{WR}$ of individual samples were calculated by summing sulfur amounts of measured AVS, CRS, and sulfate. The $\delta^{34}S_{WR}$ was calculated by measured $\delta^{34}S$ values of AVS, CRS, and sulfate in combination with their amounts. In situ sulfur isotopes of sulfides on epoxy discs were analyzed via Secondary Ionization Mass Spectrometry (SIMS) using a Cameca IMS 1280 instrument located at the Swedish Museum of Natural History, Stockholm, Sweden (NORDSIM facility)[70] for sample JTS and at the Institute of Geology and Geophysics, Chinese Academy of Sciences (IGGCAS, Beijing, China)[71] for other samples. Measurements were conducted over a rastered $10 \times 10\,\mu m$ area using a $^{133}Cs^+$ primary beam with 20 kV incident energy (10 kV primary, −10 kV secondary) and a primary beam current of ~1.0 nA. All $\delta^{34}S$ results are reported with respect to the V-CDT standard[72]. Detailed descriptions of $\delta^{34}S$ measurement parameters and standard references are given in the Supplementary Note 4. Elemental Co and Ni X-ray maps of pyrite were made in wavelength-dispersive spectrometer mode by electron microprobe (CAMECA SXFive FE) at the IGGCAS. An acceleration voltage of 20 kV, beam current of 100 nA, 3–5 μm pixel size, and dwell time of 50 ms were used. In situ trace-element analyses by laser ablation inductively-coupled plasma mass spectrometry of sulfides were made on thin sections in the GeoZentrum Nordbayern of the University Erlangen–Nürnberg, Erlangen, Germany.

**DEW calculation of sulfur concentration in fluids.** The DEW model[31,32] enables the calculation of reaction equilibrium constants involving minerals, aqueous inorganic and organic ions, complexes, and neutral species. These equilibrium constants combined with the EQ3 fluid speciation code[73] can be used to develop an aqueous speciation and solubility model at high-pressure and temperature conditions. The DEW model has been successfully applied to predict organic species, diamond formation, and nitrogen cycling in subduction zones[74,75]. EQ3 computes the equilibrium aqueous speciation of a fluid in equilibrium with certain mineral assemblages at specified temperature, pressure, and oxygen fugacity. We calculated sulfur concentrations and speciation (Fig. 6a, Supplementary Fig. 2) in slab fluids at different P–T conditions along a typical subduction geothermal gradient[36], modeling cases at 60 km (2 GPa, 400 °C), 75 km (2.5 GPa, 550 °C), 90 km (3 GPa, 700 °C), 120 km (4 GPa, 770 °C), and 150 km (5 GPa, 800 °C). The $fO_2$ of subducted oceanic crust decreases with increasing depths and increasing capacity of $Fe^{3+}$ in garnet and pyroxene[17]. Considering mineral assemblages in subduction zone rocks[17,76], $fO_2$ was set at ΔFMQ + 1 (30 km), ΔFMQ (60 and 75 km), ΔFMQ-1 (90 km), ΔFMQ-2 (120 km), and ΔFMQ-3 (150 km) for subducted oceanic crust (garnet + clinopyroxene + pyrite/pyrrhotite ± lawsonite/kyanite ± carbonate/graphite ± quartz/coesite) and overlying metasediment (muscovite + quartz/coesite + pyrite/pyrrhotite ± chlorite ± paragonite ± talc ± garnet ± clinopyroxene ± lawsonite/kyanite ± carbonate/graphite). For slab serpentinite (antigorite + orthopyroxene + pyrite/pyrrhotite ± magnetite ± olivine ± talc ± magnesite), $fO_2$ was set at ΔFMQ + 1 (30, 60 and 75 km), ΔFMQ (90 and 120 km), and ΔFMQ-1 (150 km). In order to test calculation sensitivity and improve the robustness, we calculated the results at ±1 $fO_2$ unit for every case. Aqueous sulfur species are in equilibrium with pyrite or pyrrhotite based on $fO_2$. Aqueous sulfur species considered here are $H_2S_{(g)}$, $H_2S_{(aq)}$, $HS^-$, $HSO_3^-$, $SO_3^{2-}$, $HSO_4^-$, $SO_4^{2-}$, $CaSO_4^0$, $MgSO_4^0$, $KSO_4^-$, $NaSO_4^-$, $S_3^-$, $SO_{2(g)}$ and $SO_{2(aq)}$, and the corresponding thermodynamic data are reported in the DEW 2019 spreadsheet[32].

**Effect of sulfur isotope fractionation.** Sulfur isotope fractionation of pyrite during hydrothermal processes was contoured on log $fO_2$–pH diagrams based on the method of Ohmoto[42]. These calculations monitor sulfur isotope fractionation at prograde P–T stages at depths 30 km (200 °C, 1 GPa), 60 km (400 °C, 2 GPa), 75 km (550 °C, 2.5 GPa), 90 km (700 °C, 3 GPa) and 120 km (770 °C, 4 GPa). Reactions and equations of sulfur species include $H_2S_{(aq)}$, $HS^-$, $HSO_4^-$, and $SO_4^{2-}$, and the relative isotopic fractionation ($\Delta_i = {}^{34}S_i - {}^{34}S_{H2S}$) for sulfur species were calculated at higher temperatures according to the equation[40]:

$$1000 \ln(\alpha) = \frac{a \cdot 10^6}{T^2} + \frac{b \cdot 10^6}{T} + c$$

where $a$, $b$, and $c$ are empirically-determined constants, and T is temperature in Kelvin. The equilibrium constants for reactions and activity coefficients of aqueous species were recalculated for higher P–T conditions based on the DEW model[31,32]. The abundance of sulfur species and contours of sulfur isotope fractionation (compared to initial sulfur isotope of hydrothermal fluid at $\delta^{34}S_{\Sigma S} = 0$‰) as functions of $fO_2$ and pH were calculated using Eqs. (17–25) listed in ref. [42].

The changes in isotope fractionation during pyrite crystallization from slab fluids ($H_2S$ dominated or $SO_4^{2-}$-dominated) in closed system and open-system (Rayleigh fractionation) processes were modeled at the vein-formation temperature 550 °C. In a hydrothermal system, the isotope composition of an instantaneously separated solid phase i from a fluid is:

$$\delta^{34}S_i = \delta^{34}S_0 - (1 - F) \cdot 1000 \cdot \ln(\alpha) \quad \text{(closed system)}$$

or

$$\delta^{34}S_i = \delta^{34}S_0 + 1000 \cdot \left( F^{(\alpha-1)} - 1 \right) \quad \text{(open system)}$$

where $\delta^{34}S_0$ is the initial fluid isotope composition (set as 0‰) and F is the fraction of sulfur remaining in the fluid.

**Estimate of global sulfur input into subduction zones.** The input sulfur flux $F_S$ and its isotope composition into subduction zones are:

$$F_S = \sum (L \cdot R \cdot t \cdot \rho \cdot C_S)$$

$$\text{and } \delta^{34}S = \sum \left( \delta^{34}S_t \cdot L \cdot R \cdot t \cdot \rho \cdot C_S \right) / \sum (L \cdot R \cdot t \cdot \rho \cdot C_S)$$

where L is the global length of subduction zones, R is the convergence rate, t is the thickness of the sequence layers in the slab, ρ is the density of the sequence layers, $C_S$ is the sulfur concentration [S], and $\delta^{34}S_t$ is the sulfur isotope composition of the sequence layers. The total effective length of subduction zones is ~38,500 km, which covers more than 90% of global trench length[49]. The convergence rate of 6.2 cm $yr^{-1}$ used here is taken from an average rate of 17 active oceanic subduction zones[36]. Based on the oceanic lithosphere stratigraphy (Penrose style) and its average [S] and $\delta^{34}S$ composition from the best current understanding (Supplementary Fig. 3), the calculated global sulfur input via subducting slabs is estimated to be 46.5 × $10^{12}$ g $yr^{-1}$. The bulk slab sulfur isotope composition of this sulfur input is estimated at −3.60‰. Using the same method, the calculated global water flux ($1.06 \times 10^{15}$ g $yr^{-1}$) of subducted slabs is very close to previous estimates ($1.0 \times 10^{15}$ g $yr^{-1}$)[49].

**Sulfur output and net $\delta^{34}S$ released by slab fluids.** The output sulfur flux released from the subducted slab via fluids ($f_S$) is:

$$f_S = C_{S-fluid} \cdot f_{fluid}$$

and the net sulfur isotope composition of fluids released from the subducted slab ($\delta^{34}S_{net}$) is:

$$\delta^{34}S_{net} = \sum \left( \delta^{34}S \cdot f_S \right) / \sum f_S.$$

where $C_{S-fluid}$ refers to $[S]_{fluid}$ and $f_{fluid}$ to the fluid flux released from the subducting slab. Based on water flux ($0.32 \times 10^{15}$ g$yr^{-1}$) and $[S]_{fluid}$ from the DEW model, the calculated sulfur output at 70–100 km is 2.46 × $10^{12}$ g $yr^{-1}$ (5.3% of total input $F_S$) with a $\delta^{34}S$ value of −1.84 ± 3 ‰. The net $\delta^{34}S$ value of slab fluids released at 70–100 km depths is further adjusted to −2.54 ± 3 ‰ (Fig. 9) considering fluid–rock isotopic exchange.

Our calculations indicate that along the subduction thermal gradient, at different subduction depths, the variations of temperature, pressure, $fO_2$ and pH will not cause large sulfur isotope fractionation (Fig. 7). Thus, the fluid $\delta^{34}S$ compositions obtained at 70–100 km depths can be extrapolated to different depths in the subduction zone. Following the similar assumptions and calculation approach, we obtained sulfur outfluxes and associated $\delta^{34}S$ values of slab fluids released at 30–50 km (0.00004 × $10^{12}$ g $yr^{-1}$, −1.0‰), 50–70 km (0.009 × $10^{12}$ g $yr^{-1}$, −3.2‰), 100–150 km (0.32 × $10^{12}$ g $yr^{-1}$, −0.1‰), and 150–230 km (0.11 × $10^{12}$ g $yr^{-1}$, +1.0‰), based on the water flux released from the slab at different depths as calculated by van Keken et al.[49]. The total sulfur output at 30–230 km is calculated at 2.91 × $10^{12}$ g $yr^{-1}$ (6.3% of total input $F_S$) with a $\delta^{34}S$ value of −2.13‰ (Fig. 9).

**Uncertainties on output sulfur $\delta^{34}S$ estimates.** The estimates of sulfur fluxes released from the slab have significant uncertainties. However, our study provides a robust isotopic signature for the slab fluids. The $\delta^{34}S$ estimate remains at slightly negative values in all of the following scenarios:

The uncertainty of $\delta^{34}S_{net}$ is mostly dependent on the $\delta^{34}S$ value of fluids released by the AOC at 70–100 km, which provides the major fluid flux and has a relatively high sulfur concentration (0.74 wt.%). Although we consider a large $\delta^{34}S$ range of fluid$_{AOC}$ (−6 to +4‰), the errors on the $\delta^{34}S$ of slab fluid released at 70–100 km are all less than ±2.5‰ (2σ). Changes in other parameters and assumptions cause variations of less than ±1‰ in $\delta^{34}S$. Hence, we estimate ±3‰ as a reasonable uncertainty.

Our study is based on the best current knowledge of slab structure and water budget[49]. However, new research based on ocean-bottom seismic data reports that

mantle hydration may extend up to 24 km beneath the Moho[54], which indicates that the subducting plate may contain much more water than previously thought[49]. If we adopt this assumption of a thicker serpentinized upper mantle[54] and recalculate the water and sulfur fluxes (i.e., enlarged the serpentinite-dehydrated water amounts in the subduction zone accordingly), the sulfur input increases to $7.6 \times 10^{13}$ g yr$^{-1}$ and sulfur output increases to $3.93 \times 10^{12}$ g yr$^{-1}$ but the sulfur productivity (5.2%) of the subducting slab shows little variation. More importantly, the net δ$^{34}$S displays almost no change at 70–100 km (−2.4‰). This consolidates our prediction of slab-released sulfur regarding the δ$^{34}$S signatures of arc settings.

Subducted sediment types and their redox state may have a potential effect on our results, even though there currently is no firm consensus about how much sediment is subducted. Metasediment in the Franciscan complex contains red ferruginous chert, but its proportion is subordinate compared to greywackes[77]. Moreover, at the major sulfur release window (70–100 km), the sediment contribution to the total sulfur loss is small (less than 20%). The channelized fluids[24,25] in subducted slabs (instead of pervasive fluids) prevent intensive sediment contamination of deeply-derived fluids. Here we conclude that the variable sediment protoliths and redox state will not influence our results significantly.

## Data availability
The source data underlying Figs. 3–9 and Supplementary Figs. 1 and 2 are provided as Supplementary Data.

## Code availability
The computer code including the Deep Earth Water model (2019) and EQ3 packages to perform the DEW calculations in this study is publicly available from the DEEP CARBON OBSERVATORY website [www.dewcommunity.org/resources.html] for research purposes.

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

## Acknowledgements

This project was supported by the National Key R&D Program of China (2018YFA0702701), National Natural Science Foundation of China (41772056 and 41390445). J.L.L. thanks the funding from Youth Innovation Promotion Association CAS (2018090) and the CSC for supporting his one-year stay at Yale University and three months at Freie Universität Berlin. J.J.A. gratefully acknowledges support from the U.S. National Science Foundation (EAR–1650329). NordSIMS is a Swedish infrastructure supported under VR grant 2017-00671; this is contribution 610. We thank U. Wiechert and F. Schmid for help with bulk-rock S analyses; Q. Mao and D. Zhang for help with EMP analyses; and H. Jeon, L. Chen, L.L. Dong, and J. Li for help with SIMS analyses. We are grateful to J.A.D. Connolly, S. Tassara, B.T. Li, and X.Q. Zhou for their helpful discussions and suggestions. Special thanks go to P. van Keken for providing detailed global water flux data, which enhances the reliability of our mass-balance calculation significantly.

## Author contributions

J.L.L. and T.J. designed the study. J.L.L., T.J., J.G. and R.K. collected the samples. J.L.L., R. K. and X.S.W. performed the microprobe mapping and laser trace-element analysis. E.M. S. and J.L.L. conducted the bulk sulfur isotope analysis. J.L.L. and M.J.W. conducted the SIMS analysis. F.H. performed the DEW calculation. J.L.L., E.M.S., T.J. and J.J.A. performed the mass-balance calculations. All authors contributed to the extensive discussion and manuscript writing.

## Competing interests

The authors declare no competing interests.
