## [Peer Review File · Nature Communications]

Reviewers' comments:

Reviewer #1 (Remarks to the Author):

Review of Li et al.

The MS of Li et al. uses high-pressure (HP) exhumed subduction zone rocks and enclosed veins to constrain sulfur cycling in subduction zones. They find that, despite arc volcanic rocks having high S content and positive S isotopic values, there is only modest slab-to-wedge S transfer. Using their data and modeling, Li et al. conclude that only up to 20% of subducted S is released by slab fluids along the subduction interface, and that the net isotopic composition is more negative than arc volcanic rocks. Thus, slab-derived fluid is not a major contributor of sulfate to oxidize the mantle wedge, and therefore cannot be the source of heavy $d_{34}S$ in arc volcanic rocks.

Overall, this paper presents a high-quality and detailed dataset, and the conclusions are important and timely for the subduction zone/geochemical cycling community, as well as the field of geoscience as a whole. I think the paper is suitable for publication in a high-impact journal like Nature Communications, following some minor-major revisions as well as reorganization/restructuring of parts, particularly figures. Below are my main comments:

- Lines 138-140: "Any potential effects of S isotope fractionation during hydrothermal processes should be ruled out prior to tracing the S sources..." This statement is important and central to the rationale for using exhumed HP terranes, especially in the context of studying a highly mobile element like S. However, the authors push the main discussion of this into the Supplementary Information. I think that discussion of this issue needs to be in the main text, to convince the reader that S isotope fractionation is indeed minimal for such samples. In the paragraph devoted to this the authors only cite one specific PT condition (550 C, 2.4 GPa) where they show minimal S isotope fractionation, and yet the study is focusing on the entire subduction zone interface. Furthermore, why are the calculations only done at low f_{O_2} , when prevailing views suggest that subduction zones are more oxidized than other tectonic settings? (this also seems to contradict the final conclusion of the paper that high f_{O_2} is needed somehow in the mantle wedge...)
- Line 178-179: "This influx is slightly lower than previous estimates" – how much lower, and is it significant? (one has to go all the way to the end of the SI to find out..).
- Lines 188-189: "indicating a sulfur release pulse at ~90 km depth". Is this pulse also related to the main H₂O release pulse inferred at similar depths (i.e. amphibole breakdown)? This is important because later in the paper (Lines 223-225) Li et al. invoke slab-derived H₂O to explain the high f_{O_2} of subarc mantle. However this is contentious as H₂O alone is a poor oxidizer (e.g. Frost and Ballhaus, (1998)). So this needs to be sorted out (it is confusing as written) Furthermore, what is the predicted $d_{34}S$ of this S coming off the slab at ~90 km depth?
- Lines 237-241: "Therefore, the negative-to-positive shift of $d_{34}S$ in the mantle wedge must happen during partial melting processes... under rather high f_{O_2} conditions." I find this an unsatisfactory explanation, as there is a continuing debate about the f_{O_2} of the mantle wedge, especially since most of the community has relied on melt inclusions and arc volcanic rocks to constrain f_{O_2} of the mantle (for a recent review of the debate and a new perspective on using primitive arc cumulates, see Chin et al. (2018)). So only assuming melting may occur under high f_{O_2} conditions (also, it is not specified what is "high" here) is only one possible scenario (i.e., what if melting is under low f_{O_2}).

Comments about figures:

- In general, there are two issues with the figures:
- 1) I think that some of the figures in the SI are too important to be put there.
- 2) The figures in the main text are somewhat repetitive (do you really need 3 figures with multiple panels showing detailed microphotographs and outcrop photos? This is obviously important and clearly shows that careful, detailed work was done, but I would select the most

interesting and representative examples for the main text, and move the rest to the SI), and free up space for the important stuff stuck in the SI

-

- The thermodynamic modeling of S isotope fractionation shown in Fig S2 is important. However why is only one PT condition explored (also there is no reference as to why the veins formed at 550 C, is this based on mineral thermobarometry?). What would be more valuable is the fractionation calculated along specific points in the subduction zone interface (from shallow to deep).

-

- Figure S1 is also important, it could be added to the main text and the associated writeup of the data could be shortened.

-

- Fig 1 in main text: are all 6 of these photographs necessary? Perhaps select a few key ones and combine with some of the more important ones from Fig. 3 and 4 into one figure.

-

- Fig. 2 in main text: this figure is extremely hard to read and complicated. I wonder if there is a way to make it more compact and accessible? Or maybe split up into two figures?

-

- Fig. 6: similar to Fig 2., this figure is really complicated and the text is extremely small. Look for a way to simplify...

-

- Fig. S5 – this figure seems also important and could be included in the main text (perhaps with the DEW calculation figures or a subset of them?)

-

- To summarize, I think this is a very nice paper, and worthy of publication in Nature Communications, following appropriate revisions outlined above.

-

- Respectfully,

- Emily J. Chin, Scripps Institution of Oceanography UC San Diego, July 13, 2019

-

- Chin, E.J., Shimizu, K., Bybee, G.M., Erdman, M.E., 2018. On the development of the calc-alkaline and tholeiitic magma series: A deep crustal cumulate perspective. *Earth and Planetary Science Letters* 482, 277-287.

- Frost, B.R., Ballhaus, C., 1998. Comment on "Constraints on the origin of the oxidation state of mantle overlying subduction zones: an example from Simcoe, Washington, USA" by AD Brandon and DS Draper. *Geochimica et Cosmochimica Acta* 62, 329-332.

-

Reviewer #2 (Remarks to the Author):

Review of NCOMMS-19-19518-T

Overall comments

Overall, I think this is an interesting manuscript (MS) that can be certainly published in a top-tier journal providing that important modifications are undertaken.

Novelty and relevance for Nature Comm:

I am not 100% sure this MS in its present form is suitable for Nature Comm, or a similar journal where the message should be crystal clear and the observations/models really novel.

For instance, when it comes to bulk rock analyses, the authors state (and demonstrate) that their results are just consistent with what has been reported earlier on similar reservoirs (see line by line comments).

Furthermore, much of the results on S fluxes, which are explained in the second part of the abstract, rely entirely on calculations detailed in the supplements. At present, it is not sufficiently clear why these new measurements make a difference in comparison with earlier reports, and how these new results have been directly used to infer S fluxes or ^{34}S signatures of slab-derived agents. From the supplements, it appears instead that S flux calculations rely on a synthesis of literature data, so you have to outline how your data is used in a much clearer way. Of course, this is understandable for a reviewer, but not for a reader.

I think a way to improve this is to put the modeling in the discussion (see below).

If this MS was to be sent for revisions by the Nature Comm editors, I would suggest a major overhaul of the presentation and re-organization to take advantage of the space given by Nature Comm. This would not only help the reader by improving the thoroughness of the presentation (intro/results/discussion, see below), but will also clarify what is really new in this paper.

Scientific merit:

The dataset is thorough and of high quality, it will definitely be of interest for the community to have these numbers out and I would like to support publication of this study, wherever it is. I have no real negative criticism on data.

There are several conceptual issues, notably regarding to which extent HP veins can be representative of their parental slab-derived agents. In general, veins can also be seen as 'cumulates'. This means that if they can indeed provide clues on the nature of their parental agents, these latter have to be calculated from the vein features. In other words, in some cases, veins can only be made of all the stuff that the agent was not able to transport further. For instance, it is possible to have a percolating SO_4 -bearing melt that fractionate sulfides in a channel because of a redox interaction with an Fe-rich host rock; the only observation you get are the Fe-rich sulfides.

SO_4 being way more soluble than S^{2-} in silicate melts and aqueous fluids, I therefore do not understand clearly why the scarcity of sulfates in devolatilized slab rocks indicates why they were not transferred to the overlying mantle wedge at different stages.

Likely the authors have data from earlier papers in hand to correct some of these points, but a summary of background data regarding the petrography (mineral assemblage for $f\text{O}_2$ conditions), P-T conditions and relationships between samples in the exhumed terrane hasn't been included in the MS (see below).

The MS lacks precise quantification of $f\text{O}_2$ (and possibly pH) to directly assess the relevance of the thermodynamic models. Petrographic observations are certainly useful, but can only be used in support of calculations. Is there any tool that can provide these numbers?

Quality of the text and presentation:

The MS is generally well-written and to the point, with a good English style, and I enjoyed reading it.

There are several shortcuts that must be corrected prior to publication. These are not related to the English style, but the text appears sometimes written in a pre-oriented way highlighting the novelty of the hypothesis the authors support (e.g. in the introduction to present the scientific problem). Sometimes, the authors oversell their findings, leaving sometimes the impression that they are the first to report really relevant S data on HP slab rocks. This has to be thoroughly corrected.

I tend to disagree with the bold statements that the authors put forward in their abstract to sell their research. I think that the problem is way more complex than how it is presented here, both in terms of what is currently known (all recent Evans et al. papers on sulfides in serpentinite-eclogites) and what has to be solved (see line by line comments). This has to be reworked with the space limits allowed by Nature Comm.

For instance, a problem is how the interplay between redox and degassing is laid out in the introduction, and how it is intended to present the scientific context and outline the importance of this study. You need to be more specific and accurate (see line by line comments).

I am not sure as well that important background details on the samples are enough visible and comprehensible instantly to provide a clear view to the reader from the beginning. This is key for a Nature paper in my opinion. In particular, Nature Comm can accept much longer papers than other Nature journals, and I think the authors have not fully taken advantage of this. Therefore, some aspects of the study (rationale in terms of sample choice) remain really obscure at an early stage (e.g., difference in P-T conditions for bulk rocks and vein1 and 2; see line by line comments). If Nature Comm allows this, I think these information and others can be summarized in a 'background data' part of 10-15 lines just after the introduction, i.e. rather than being partially presented with the 34S results, or sometimes very lately in the discussion (see line by line comments). Important information on samples is somehow completely hidden in the middle of other things, this has to be re-organised.

Regarding modeling, there are too many links that have to be done by the reader himself between what is presented in the last part of the results and calculation procedures, which massively rely on the supplements. This further contributes to the flawed aspect of the presentation. Many of the statements on modeling results in the main text become cryptic for a first-time reader because of this. Re-writing should be performed here as well.

Your discussion is actually very short and only speculative, I would suggest transferring much of the modeling information in this section (see above). Speculations on future research paths can be placed in such a journal, but they have to be limited 5-10 lines at the very end of the paper. Why not discussing modelling results with greater and clearer details (and a re-organisation) in the discussion instead?

Your data should be compared to recent papers by the Evans group that you have cited. These have evidenced high 34S in eclogites, and one of the hypotheses to explain this is that mantle serpentinite dehydration from below the eclogitic crust continues to greater depths. Looking at your Fig. S1, adding these data would considerably change the picture. After all, you provide data on only one locality and the availability of oxidized and high 34S from serpentinites below may also change.

Figures are large and complex but clear and comprehensible, minor modifications need to be undertaken, such as symbols in Fig. 2c, which variable shape can be misleading. Symbols must be defined in each figure caption as well.

Line by line comments of the main text and figures

Line 32-33: does this observation imply that sulfate transported by the slab is low or that it was lost during devolatilization?

Line 47-50: here you need to state that this putative high-34S addition is under the form of sulfate. Otherwise you wouldn't have such high S contents (up to 3000 ppm), which correspond to the S content at sulfate saturation in andesitic melts (Botcharnikov et al. 2011).

Line 50-51: there are no such alternative models, this is actually the kind of model you propose in this MS. This statement (with no references) does not describe the following lines; i.e. suggesting

that degassing can influence ^{34}S does not mean that (oxidized) S hasn't been added from the slab.

Degassing can increase ^{34}S , true, but first, the Mandeville's inclusions contain 300 ppm S at most. More recent papers by Fiege et al. (2014, 2015) have shown that the fractionation factors between melt and fluid can be also highly dependent on redox state, especially above the sulfide/sulfate transition at FMQ+1. Producing high ^{34}S in melts can be done at these conditions, which is actually very close to what is expected for primitive arc compositions (e.g. peridotites, Bénard et al. 2018 in Chem Geol).

The high sulfur in many arc inclusions has to come from somewhere, and undegassed (sulfate-saturated), high- ^{34}S and S (3000 ppm or more) sub-arc mantle inclusions have trace element spikes typical of slab agents (e.g. Bénard et al. 2018 in Nature Comm).

The real challenge of this research is somehow twisted with lines 50-54 coz ref. 10 says there is no sulfate in primary arc magmas, whereas only sulfate-present degassing can produce increases in ^{34}S significant enough to reproduce arc inclusions. If it is not oxidized, this means that ^{34}S is in the source as part of a recycled component or produced by the melting process.

Line 53-54: same as above, this is not the debate that is currently in the community. In a Nature paper you need to carefully lay out the scientific problem, please re-write this paragraph accordingly.

Line 65-69: are these pseudotachylite veins? Are they quenched? If not, you are probably overstating the importance of HP veins in directly reflecting the nature of the agents from which they formed (see overall comments).

Line 73-74: 'unparalleled' is really daring here, earlier studies have also looked at HP rocks and their sulfides...

Line 77: what do you mean exactly by unaltered, that the sulfides present fresh primary textures? For some people, serpentinization means alteration, so you have to be more specific here, or alternatively, you could remove this word to later evidence precisely the degree of alteration of each phase in your petrographic description.

Line 77-92: this part is well-written and clear in terms of ^{34}S results, but on the contrary to the following one on veins, it provides very little background information on samples. Yet, these can be really important at this stage to understand your objectives with these analyses. I am not talking about details and repeating what has been already published in ref. 27, but rather, I would have expected a short explanation of field relationships, and more importantly, thermobarometric estimates for metabasites, metapelites and serpentinites. How the reader is supposed to understand what these numbers mean if no information is provided about the devolatilization sequence?

Line 88: well, in that case this is not really suitable for a Nature paper. Furthermore, this is not entirely true coz Evans et al. (2014) have reported high ^{34}S in eclogites...

Line 112-114: this rather looks like interpretation, which must be transferred to the very beginning of the discussion.

Line 115-117: thermobarometric estimates should be specified for these rocks as well.

Line 146: an example of why adding more petrological background would be important, as these P-T values have to be checked in comparison with the formation conditions of all your samples. If the P-T conditions are the same for all samples (as mentioned very shortly, hidden to be honest, in the supplements), this has to be specified way more clearly as well.

Line 150-154: again, this means the results are not that novel.

Line 158-160: where and how is this interpreted? In your paper? Based on the literature using other geochemical proxies? Please re-write.

Line 168-170: I didn't get the full logic here, please re-write and give more details. Do you mean that residual Co is stored in serpentinite sulfides after SO₄ flushing, and that it is thus normal to not have in in great quantities in the metasomatic pyrite? If so, please re-write accordingly.

Line 173-174: can you be more specific and show what you are talking about, instead of just stating that things are consistent with your interpretation? Again, a preliminary petrological background part would be useful here, as it would prevent the appearance of new petrological observations (for the reader at least) in an odd manner in the results.

Line 195: ok, this is where your data are important, crucial to understand the rationale of your study, please outline this way more clearly here and elsewhere in the MS when re-organizing it.

Line 209-245: this is not really a discussion and rather a list of speculations. Since a model cannot really be considered as the central result of your study (these are the measurements right?), I would suggest transferring the models in the discussion.

The speculations are fine for the most part, but it should be condensed to the minimum and not constitute the core of what is called 'discussion'.

However, some aspects of your speculations have been heavily contradicted in the literature, such as the potential of H₂O as an oxidizer...

Line 219-220: this observation should appear way before this part, such as a preliminary background data part (see above). Otherwise, this is disorganized.

Line 560-561 and 570-571: these statements are obscure, it is impossible to make the link between what is shown in the figure and these conclusions without explanations. Wouldn't it be better to put clearer sentences but only in the discussion part?

Line 595-603: would be nice to see where your samples are in this diagram, in particular what they do represent in terms of P-T conditions and S cycle. This would nicely complement the background data text part I suggest.

Line by line comments of the supplements

Line 53-70: why not reporting the ³⁴S data of Evans et al. recent papers on the Alps, New Caledonia and Corsica? Another question is what are the rocks that are not shaded in grey? Are these all chlorite harzburgite formed from antigorite breakdown?

Line 95: what do you mean by 'Fe-saturated' in this context? Fe-rich?

Response to Reviewer #1 (Emily J. Chin):

Reviewer #1 (Remarks to the Author):

Review of Li et al.

The MS of Li et al. uses high-pressure (HP) exhumed subduction zone rocks and enclosed veins to constrain sulfur cycling in subduction zones. They find that, despite arc volcanic rocks having high S content and positive S isotopic values, there is only modest slab-to-wedge S transfer. Using their data and modeling, Li et al. conclude that only up to 20% of subducted S is released by slab fluids along the subduction interface, and that the net isotopic composition is more negative than arc volcanic rocks. Thus, slab-derived fluid is not a major contributor of sulfate to oxidize the mantle wedge, and therefore cannot be the source of heavy $\delta^{34}\text{S}$ in arc volcanic rocks.

Overall, this paper presents a high-quality and detailed dataset, and the conclusions are important and timely for the subduction zone/geochemical cycling community, as well as the field of geoscience as a whole. I think the paper is suitable for publication in a high-impact journal like Nature Communications, following some minor-major revisions as well as reorganization/restructuring of parts, particularly figures.

Response: We thank the reviewer for her positive and encouraging comments.

Below are my main comments:

- Lines 138-140: “Any potential effects of S isotope fractionation during hydrothermal processes should be ruled out prior to tracing the S sources...” This statement is important and central to the rationale for using exhumed HP terranes, especially in the context of studying a highly mobile element like S. However, the authors push the main discussion of this into the Supplementary Information. I think that discussion of this issue needs to be in the main text, to convince the reader that S isotope fractionation is indeed minimal for such samples. In the paragraph devoted to this the authors only cite one specific PT condition (550 C, 2.4 GPa) where they show minimal S isotope fractionation, and yet the study is focusing on the entire subduction zone interface. Furthermore, why are the calculations only done at low $f\text{O}_2$, when prevailing views suggest that subduction zones are more oxidized than other tectonic settings? (this also seems to contradict the final conclusion of the paper that high $f\text{O}_2$ is needed somehow in the mantle wedge...)

Response: (1) We moved information from the Supplementary Material to the main text, and re-structured the whole paragraph (Result subsection #5) to make a clear statement that sulfur isotope fractionations under subduction zone P-T- $f\text{O}_2$ -pH conditions are minimal. Along with this re-organization, we wrote a paragraph to address the absence of sulfate in slab fluids (first paragraph of the Discussion), which is critical to support the sulfur isotope fractionation. Related calculation methods were combined into the Method part.

(2) We recalculated the $f\text{O}_2$ -pH diagram (Fig. 7a-d) at different P-T conditions (200°C, 1 GPa; 400°C, 2 GPa; 700°C, 3 GPa; 770°C, 4 GPa, following the typical subduction zone thermal gradient) to show limited $\delta^{34}\text{S}$ fractionation at different depths in the subduction zone, with given $f\text{O}_2$ and calculated pH conditions.

(3) Subduction zones are not more oxidized than other tectonic settings. The $f\text{O}_2$ of the asthenospheric mantle, and therefore for MORBs, is close to FMQ-1, ranging from FMQ-2 to FMQ (Bézos and Humler, 2005; Frost and McCammon, 2008). In contrast, the $f\text{O}_2$ of metasomatized peridotites from the mantle wedge above subducting slabs is more oxidized at FMQ to FMQ+2 (substantial proportion above FMQ+1) (Parkinson and Arculus, 1999; Frost and McCammon, 2008). Arc lava shows much higher $f\text{O}_2$ above FMQ+2 (Rohrbach et al., 2005; Carmichael et al., 2006). However, until now, only a few studies constrained in situ prograde $f\text{O}_2$ of a subducted slab due to its complex redox condition and evolution (Foley, 2011). Before subduction, the redox condition of oceanic crust affected by seawater alteration is modified between FMQ-1 and FMQ+1 (Foley, 2011). Once entering the subduction zone, previous studies suggest that the $f\text{O}_2$ of subducted oceanic crust decreases with increasing depth, to <FMQ buffer at the

blueschist-eclogite transition at depths of 60-80 km (Li et al., 2016; Tao et al., 2018). Serpentinized slab mantle is complicated (can be both oxidizing and reducing) due to different degrees of pre-subduction serpentinization (Evans et al., 2017). The normal slab mantle underlying the mafic oceanic crust (low water:rock ratio) is most likely partly serpentinized (e.g. Bloch et al., 2018; Ranero et al., 2003; Korenaga, 2017), and contains widespread sulfides and awaruite, indicating reduced conditions. This is why we used reduced redox conditions (low fO_2) during calculations. To illustrate this, we added a detailed discussion of the fO_2 of slab sequences in the supplementary material, and added necessary information in the main text. See related response to Rreviewer #2, and [L204-215, Pages 8-9].

- Line 178-179: “This influx is slightly lower than previous estimates” – how much lower, and is it significant? (one has to go all the way to the end of the SI to find out..).

Response: Our subduction zone sulfur input is estimated at 4.55×10^{13} g/yr. This value is lower but at the same order of magnitude with previous estimates of $6.7\text{--}8.3 \times 10^{13}$ g/yr (including $4.8\text{--}6.4 \times 10^{13}$ g/yr sulfur subducted by altered oceanic crust (Alt and Shanks, 2011), 1.3×10^{13} g/yr sulfur subducted by sediment (Canfield, 2004), and 0.64×10^{13} g/yr sulfur subducted by slab serpentinite (Alt et al., 2012)) and $6.9\text{--}11.9 \times 10^{13}$ g/yr (Evans, 2012). All the subduction sulfur input estimates have great uncertainties. We think this discrepancy is acceptable. As suggested by the reviewer, we moved Fig. S5 to the main text (now Fig. 8b). The reader can easily see the difference between our estimate with previous ones from this diagram, as also described in a revised sentence [L345-346, Page 13].

- Lines 188-189: “indicating a sulfur release pulse at ~90 km depth”. Is this pulse also related to the main H₂O release pulse inferred at similar depths (i.e. amphibole breakdown)? This is important because later in the paper (Lines 223-225) Li et al. invoke slab-derived H₂O to explain the high fO₂ of subarc mantle. However this is contentious as H₂O alone is a poor oxidizer (e.g. Frost and Ballhaus, (1998)). So this needs to be sorted out (it is confusing as written) Furthermore, what is the predicted d³⁴S of this S coming off the slab at ~90 km depth?

Response: Yes, the sulfur release pulse at ~90 km depth is consistent with the main H₂O release pulse, since the sulfur concentration in fluids reaches a peak (see DEW result Fig. 6a) at the major water release window (Fig. 6b) of the slab modeled by van Keken et al. (2011). We have shown this in the main text: “This sulfur release window (Fig. 6) coincides with pyrite-to-pyrrhotite breakdown¹² (releasing H₂S) and the major slab fluid release (~32% H₂O/H₂O_{total})...” [L365-366, Page 14]. In order to illustrate this better, we made another diagram (now Fig. 6b) to show the coincidence of sulfur and H₂O release windows from the slab.

Whether H₂O alone as an oxidizer at the mantle wedge environment remains controversial (Brandon and Draper, 1996; Frost and Ballhaus, 1998). The debate is largely dependent on the capacity and efficiency of H₂ removal from the system during H₂O dissociation (Frost and Ballhaus, 1998; Brandon and Draper, 1998). The most recent paper (Tollan and Hermann, 2019) suggests that H₂ can be incorporated into orthopyroxene of the mantle wedge, transferring the hydrous melts to oxidized conditions and thus oxidized arc magmas. Therefore, slab fluids are not necessarily oxidized and H₂O alone seem to be a potential oxidizer to arc magmas. However, this is mostly beyond the scope of this manuscript. Our point is that slab fluids cannot transfer sulfate into the mantle wedge. We rewrote this part to make it clearer, see [L417-422, Page 16].

The $\delta^{34}\text{S}$ value of slab fluids released at ~90 km depth is about -2.5‰. We did not calculate it for 90 km depth only, but for a range of 70-100 km. This is displayed in the final diagram Fig. 9.

- Lines 237-241: “Therefore, the negative-to-positive shift of d³⁴S in the mantle wedge must happen during partial melting processes... under rather high fo₂ conditions.” I find this an unsatisfactory explanation, as there is a continuing debate about the fo₂ of the mantle wedge, especially since most of the community has relied on melt inclusions and arc volcanic rocks to constrain fo₂ of the mantle (for a recent review of the debate and a new perspective on using primitive arc cumulates, see Chin et al. (2018)). So

only assuming melting may occur under high fo2 conditions (also, it is not specified what is “high” here) is only one possible scenario (i.e., what if melting is under low fo2).

Response: We rewrote this part to provide more scenarios possibly happened during partial melting processes [see L433-443, Page 17]. Since partial melting processes and oxidization within the mantle wedge are complex, more detailed discussions are beyond the scope of the present study which focuses on the fluid contribution from devolatilizing slabs.

Comments about figures:

- In general, there are two issues with the figures:

- 1) I think that some of the figures in the SI are too important to be put there.

Response: We moved Fig. S1 (now as Fig. 3), Fig. S2 (now as Fig. 7e-g), and Fig. S5 (now as Fig. 8b) into the main text, as suggested by the reviewer.

- 2) The figures in the main text are somewhat repetitive (do you really need 3 figures with multiple panels showing detailed microphotographs and outcrop photos? This is obviously important and clearly shows that careful, detailed work was done, but I would select the most interesting and representative examples for the main text, and move the rest to the SI), and free up space for the important stuff stuck in the SI

Response: We adjusted our original figures. See general response point #3 to the editor.

- The thermodynamic modeling of S isotope fractionation shown in Fig S2 is important. However why is only one PT condition explored (also there is no reference as to why the veins formed at 550 C, is this based on mineral thermobarometry?). What would be more valuable is the fractionation calculated along specific points in the subduction zone interface (from shallow to deep).

Response: We moved Fig. S2 into the main text (now as Fig. 7e-g), and did more calculations to show limited sulfur isotope fractionation during hydrothermal processes (Fig. 7a-d). See response to Main Comment #1.

- Figure S1 is also important, it could be added to the main text and the associated writeup of the data could be shortened.

Response: We moved Fig. S1 to the main text (now as Fig. 3).

- Fig 1 in main text: are all 6 of these photographs necessary? Perhaps select a few key ones and combine with some of the more important ones from Fig. 3 and 4 into one figure.

Response: We endeavored to better organize a number of the figures. We gathered the same kind of photographs of different samples in one figure. See general response point #3 to the editor

- Fig. 2 in main text: this figure is extremely hard to read and complicated. I wonder if there is a way to make it more compact and accessible? Or maybe split up into two figures?

Response: We deleted (b) from this figure and moved it to the supplementary information (now Fig. S1). The new figure (Fig. 4) is compact and clear now, as (a) shows the bulk-rock and (b) shows the in-situ sulfur isotope compositions.

- Fig. 6: similar to Fig 2., this figure is really complicated and the text is extremely small. Look for a way to simplify...

Response: This is one of the most important figures in this manuscript. Using this cartoon, we aim to get across the main points of the paper. We tried our best to simplify it, enlarge numbers, and make it clearer and more understandable to readers. We deleted some numbers from this cartoon. In addition, as suggested by Reviewer #2, we added red stars into the diagram to show where our samples were formed in the subduction zone. See the new Fig. 9.

- Fig. S5 – this figure seems also important and could be included in the main text (perhaps with the DEW calculation figures or a subset of them?)

Response: We moved Fig. S5 to the main text (now as Fig. 8b).

- To summarize, I think this is a very nice paper, and worthy of publication in Nature Communications, following appropriate revisions outlined above.

- Respectfully,

- Emily J. Chin, Scripps Institution of Oceanography UC San Diego, July 13, 2019

-

- Chin, E.J., Shimizu, K., Bybee, G.M., Erdman, M.E., 2018. On the development of the calc-alkaline and tholeiitic magma series: A deep crustal cumulate perspective. *Earth and Planetary Science Letters* 482, 277-287.

- Frost, B.R., Ballhaus, C., 1998. Comment on " Constraints on the origin of the oxidation state of mantle overlying subduction zones: an example from Simcoe, Washington, USA" by AD Brandon and DS Draper. *Geochimica et Cosmochimica Acta* 62, 329-332.

Response: We are very grateful to Professor Chin for her positive and encouraging comments. The recommended first reference was adopted, discussed, and cited in the manuscript. The second is not included, as whether H₂O alone as an oxidizer in the mantle wedge environment remains controversial. Detailed discussion is beyond the scope of this study. See the response to Main Comment of Line 188-189.

Response to Anonymous Reviewer #2:

Reviewer #2 (Remarks to the Author):

Review of NCOMMS-19-19518-T

Overall comments

Overall, I think this is an interesting manuscript (MS) that can be certainly published in a top-tier journal providing that important modifications are undertaken.

Response: Many thanks for the positive and encouraging comments. The constructive review comments have considerably improved the quality of the manuscript.

Novelty and relevance for Nature Comm:

I am not 100% sure this MS in its present form is suitable for Nature Comm, or a similar journal where the message should be crystal clear and the observations/models really novel.

For instance, when it comes to bulk rock analyses, the authors state (and demonstrate) that their results are just consistent with what has been reported earlier on similar reservoirs (see line by line comments).

Response: We point out that the bulk-rock sulfur isotope compositions of high-pressure rocks measured in our study are consistent with previous studies, which consolidates our point to build a general picture of sulfur reservoirs in slab sequences. This is not the major novelty of the manuscript. Just as Evans et al. (2014) says in their paper: "More detailed studies of the distribution of sulfides with relation to possible fluid pathways on different length scales is required to resolve... Further work is required to constrain the role of locally and externally derived fluids, and to quantify the contributions made by the different sulfur sources." In this study, our novel point is that we identified the $\delta^{34}\text{S}$ compositions of fluids derived from subducted slabs using in-situ and bulk-rock analyses of eclogite-facies veins; in addition, we calculated the sulfur concentrations in slab fluids using the newest DEW model, which play a key role in determining the slab sulfur transport. We quantify the sulfur input and output at subduction zones due to slab dehydration and re-define the subduction zone sulfur cycle.

Furthermore, much of the results on S fluxes, which are explained in the second part of the abstract, rely entirely on calculations detailed in the supplements. At present, it is not sufficiently clear why these new measurements make a difference in comparison with earlier reports, and how these new results have been directly used to infer S fluxes or 34S signatures of slab-derived agents. From the supplements, it appears instead that S flux calculations rely on a synthesis of literature data, so you have to outline how your data is used in a much clearer way. Of course, this is understandable for a reviewer, but not for a reader. I think a way to improve this is to put the modeling in the discussion (see below).

Response: We made several changes to compare our results with previous studies, introduce the background for the calculations, and better illustrate how the data is used:

- (1) For sulfur concentration constraints, we added the current knowledge and problem to show how our calculated DEW results promote progress for understanding the slab sulfur outflux [see L351-355, Page 14]
- (2) We moved Fig. S5 into the main text (now as Fig. 8b) to show the difference between our sulfur influx and outflux estimates with previous work. We also rephased the associated sentences to compare our result with previous work [L359-360, Page 14]. See also the response to Reviewer #1 comment Line 178-179.
- (3) The way the sulfur $\delta^{34}\text{S}$ in slab fluids is calculated was also rephased to show directly how our data is used [see L380-391, Page 15]

If this MS was to be sent for revisions by the Nature Comm editors, I would suggest a major overhaul of the presentation and re-organization to take advantage of the space given by Nature Comm. This would not only help the reader by improving the thoroughness of the presentation (intro/results/discussion, see below), but will also clarify what is really new in this paper.

Response: The manuscript presentation and organization were extensively re-structured. See our general response #2 to the editor.

Scientific merit:

The dataset is thorough and of high quality, it will definitely be of interest for the community to have these numbers out and I would like to support publication of this study, wherever it is. I have no real negative criticism on data.

Response: We are grateful to the reviewer for the positive and encouraging comments.

There are several conceptual issues, notably regarding to which extent HP veins can be representative of their parental slab-derived agents. In general, veins can also be seen as 'cumulates'. This means that if they can indeed provide clues on the nature of their parental agents, these latter have to be calculated from the vein features. In other words, in some cases, veins can only be made of all the stuff that the agent was not able to transport further.

For instance, it is possible to have a percolating SO₄-bearing melt that fractionate sulfides in a channel because of a redox interaction with an Fe-rich host rock; the only observation you get are the Fe-rich sulfides.

SO₄ being way more soluble than S²⁻ in silicate melts and aqueous fluids, I therefore do not understand clearly why the scarcity of sulfates in devolatilized slab rocks indicates why they were not transferred to the overlying mantle wedge at different stages.

Response: Treating veins strictly as “cumulates” is inappropriate. For an open system, it is not the case that the vein fluids could not transport anything “further”; in fact, they transport constituents all along the conduits. Veins are mineralized fractures. Vein geochemistry may not exactly represent the fluid compositions, but has been in equilibrium with the fluid while the vein minerals formed or equilibrated with the vein fluid during recrystallization processes (e.g., Ague, 2011; Gao et al., 2007; John et al., 2004, 2008, 2012). The mineralogy in the vein and surrounding altered selvages provides valuable information regarding transported fluid constituents and the fluid-rock reactions that took place along flow paths. Veins provide a unique perspective in this regard.

Regarding sulfate, we list several arguments to illustrate why we propose little or no sulfate in fluids (the key arguments have been added into the main text at the beginning of the Discussion):

- (1) In eclogites and blueschists, there is not much hard evidence that sulfate has once been a widely stable phase under eclogite-facies conditions and all was dissolved into fluids and transported away during subduction. If sulfate was once widely distributed in these rocks, we should find relics in the matrix or as inclusions preserved in other minerals like garnet. SO₄²⁻ is indeed more soluble than reduced sulfur phases like H₂S and HS⁻ in fluids. If slab fluids contain a significant amount of SO₄²⁻ or are SO₄²⁻-dominated, at least some sulfate relics should be observed in the vein minerals. However, our comprehensive petrological study did not reveal them (except those in late-stage fractures).
- (2) If sulfate is the dominant sulfur species in fluids, it would have to get transported into the selvage as well due to the chemical potential gradient (we do observe a [S] increase trend from host rock toward the vein). This alteration should be recorded by the reaction halo as we show in the vein samples. However, no sulfate or its related oxidized chemical gradient was found by microscopy or from bulk-rock sulfur analyses. Reduction from S⁶⁺ to S²⁻ will cause oxidation in the form of hematite or magnetite, which are absent in veins or selvages. In addition, the sulfur isotope values of the produced sulfides should have very heavy δ³⁴S values (if the reaction approaches completion), which are also not detected in vein samples.
- (3) We agree that a certain amount of sulfur in the form of sulfate has been transported into the subduction zone by sediments, altered oceanic crust and serpentinite, as a result of pre-subduction hydrothermal seafloor alteration. But we argue that most of the sulfate in sediments and AOC has been dissolved into fluids and was lost at early stages of subduction, for example, at fore-arc depths as Lee et al. (2018) suggested. The fluids at shallow levels (<40km) should contain a certain amount of sulfate. This is also supported by the high oxygen fugacity of retrograde fluids, pyrite overprinted by magnetite, and albite-sulfate-magnetite-bearing fracture infillings recorded in exhumed eclogites (Li et al., 2016).
- (4) Walters et al. (2019) suggested that sulfates are likely the dominant sulfur species in slab-derived fluids due to the large δ³⁴S variation (~36‰) of the metasomatic sulfides they studied. We do not agree with this interpretation because:
 - a) They did not find any sulfate in their HP rocks and do not have any petrological evidence for the presence of sulfate. This is a major concern.
 - b) The proposed CaSO₄-producing reaction during prograde metamorphism $7 \text{Fe}_2\text{O}_3 \text{ (in silicates or oxides)} + \text{FeS}_2 + \text{CaO (in silicate)} = 15 \text{FeO (in silicates)} + 2\text{CaSO}_4$ will not happen in actual subduction environments, because during eclogitization, growth of both garnet (Fe³⁺/ΣFe up to 7%) and omphacite (Fe³⁺/ΣFe up to 50-60%) likely take all Fe³⁺ from the system, which causes the *f*O₂ to decrease gradually with increasing depth (Li et al., 2016, Tao et al., 2018). Consequently, there is little or no Fe³⁺ to oxidize sulfides. This is why diamond or graphite can be stable at great depths in subduction zones (Frost and McCammon, 2008).
 - c) The large range of δ³⁴S (~36‰) of the metasomatic sulfides is straightforward to explain. Most of the metasomatic sulfides studied by Walter et al. (2019) are formed during retrograde fluid overprinting (see Table 1 in Walter et al. 2019). The retrograde fluids are relatively oxidized, as we discussed earlier. Under high *f*O₂ conditions, isotope fractionation is large and can produce large

$\delta^{34}\text{S}$ variations. When Walters et al. (2019) argue that sulfates are likely the dominant sulfur species in fluids, they are referring to the retrograde fluids which have no direct relation to prograde slab-dehydration related fluids and arc magmatism. These oxidizing fluids at shallow depths (which may contain significant sulfate) may overprint eclogites and blueschists during their exhumation along the plate interface, but during intraslab-dehydration and fluid-flow processes, especially at great depth (>60 km), it is apparently not the case.

- (5) Debret and Sverjensky (2017) used a theoretical chemical mass transfer calculation to predict highly oxidizing sulfate-bearing fluids near the HM buffer generated during serpentinite breakdown. However, they set the initial $f\text{O}_2$ at a very oxidizing condition (FMQ+4.2 at 650°C and 2.0GPa), which is unlikely for a normal slab serpentinite. The presence of sulfide minerals can decrease the fluid $f\text{O}_2$ to FMQ+2 (Debret and Sverjensky, 2017). In addition, for incompletely serpentinitized rocks, the presence of Fe-Ni alloys can further buffer the fluid $f\text{O}_2$ at even lower values (Evans et al., 2017). We argue that the fluids derived from “regular” and “standard” serpentinite dehydration may have higher $f\text{O}_2$ values than AOC, but should be less than FMQ+2. This means that fluids from serpentinite may contain sulfate, but this should not be a predominant species in comparison to reduced species.

Likely the authors have data from earlier papers in hand to correct some of these points, but a summary of background data regarding the petrography (mineral assemblage for $f\text{O}_2$ conditions), P-T conditions and relationships between samples in the exhumed terrane hasn't been included in the MS (see below).

Response: As suggested by the reviewer, we added a section named “Sample Background” at the beginning of the Results. See our response point #1 to editor, as well as [L92-132, Pages 4-5] in the manuscript.

The MS lacks precise quantification of $f\text{O}_2$ (and possibly pH) to directly assess the relevance of the thermodynamic models. Petrographic observations are certainly useful, but can only be used in support of calculations. Is there any tool that can provide these numbers?

Response: Yes, precise quantifications of $f\text{O}_2$ and pH are critical for our thermodynamic model. However, to the best of our knowledge, it is difficult to constrain $f\text{O}_2$ and its evolution within the subduction zone through calculation/modeling. Here we mostly rely on the petrographic observations to estimate $f\text{O}_2$ conditions, and results from previous studies. We argue that the geological evidence is more robust than calculations, if no reliable theoretical means can be used. In order to better illustrate the state-of-the-art knowledge on subduction oxygen fugacity and the range of $f\text{O}_2$ values of the slab lithologies used in the models, we provide a detailed discussion in the supplementary information. We hope that this discussion is sufficient for the reader to understand $f\text{O}_2$ conditions in subduction zones. Key information was also given in the main text. With given $f\text{O}_2$ conditions we calculated the related pH values of the subduction zone fluids via the DEW model.

Quality of the text and presentation:

The MS is generally well-written and to the point, with a good English style, and I enjoyed reading it.

There are several shortcuts that must be corrected prior to publication. These are not related to the English style, but the text appears sometimes written in a pre-oriented way highlighting the novelty of the hypothesis the authors support (e.g. in the introduction to present the scientific problem). Sometimes, the authors oversell their findings, leaving sometimes the impression that they are the first to report really relevant S data on HP slab rocks. This has to be thoroughly corrected.

Response: We corrected or toned down all the perhaps overenthusiastic descriptions that the reviewer pointed out. For example:

“Although sulfide phases and their in-situ $\delta^{34}\text{S}$ compositions from high-pressure (HP) rocks have been reported in recent studies¹³⁻¹⁵, the speciation, flux, and isotopic composition of sulfur in slab fluids remain unclear.” [L63-65, Page 3]

I tend to disagree with the bold statements that the authors put forward in their abstract to sell their research. I think that the problem is way more complex than how it is presented here, both in terms of what is currently known (all recent Evans et al. papers on sulfides in serpentinite-eclogites) and what has to be solved (see line by line comments). This has to be reworked with the space limits allowed by Nature Comm.

Response: Considering the word limits, we removed the text the reviewer had concerns about, and instead keep the focus on the scientific background and current outstanding questions [see L25-29, Page 2].

Regarding what is currently known, we compare our results to recent studied by Evans et al. and their work on sulfides in serpentinites and eclogites in the Results: “ $\delta^{34}\text{S}$ values of fluids from different slab reservoirs” [see L273-278, Page 11 and L286-290, Page 11].

For instance, a problem is how the interplay between redox and degassing is laid out in the introduction, and how it is intended to present the scientific context and outline the importance of this study. You need to be more specific and accurate (see line by line comments).

Response: The degassing case was removed from the beginning of the introduction since degassing will not cause large sulfur isotope fractionation of arc melts (Fiege et al., 2014, 2015). We focus on the specific scientific problems that are currently of great concern to the community and endeavor to present it as accurately as possible. For details see the response to Comment Line 50-51 of Reviewer #2.

I am not sure as well that important background details on the samples are enough visible and comprehensible instantly to provide a clear view to the reader from the beginning. This is key for a Nature paper in my opinion. In particular, Nature Comm can accept much longer papers than other Nature journals, and I think the authors have not fully taken advantage of this. Therefore, some aspects of the study (rationale in terms of sample choice) remain really obscure at an early stage (e.g., difference in P-T conditions for bulk rocks and vein1 and 2; see line by line comments). If Nature Comm allows this, I think these information and others can be summarized in a ‘background data’ part of 10-15 lines just after the introduction, i.e. rather than being partially presented with the ^{34}S results, or sometimes very lately in the discussion (see line by line comments). Important information on samples is somehow completely hidden in the middle of other things, this has to be re-organised.

Response: As suggested by the reviewer, we added a subsection “Sample Background” at the beginning of the Results. See our general response point #1 to editor.

Regarding modeling, there are too many links that have to be done by the reader himself between what is presented in the last part of the results and calculation procedures, which massively rely on the supplements. This further contributes to the flawed aspect of the presentation. Many of the statements on modeling results in the main text become cryptic for a first-time reader because of this. Re-writing should be performed here as well.

Response: We moved the necessary information regarding modeling from the supplementary material to the main text. For example, we give the background (e.g., P-T conditions, $f\text{O}_2$ estimates) for DEW calculations, and evaluated the effects to the results if $f\text{O}_2$ changes at a certain range [see L216-219, L227-229, Page 9]. We introduced important parameters (e.g., [S], $\delta^{34}\text{S}$, thickness, and density of oceanic lithosphere stratigraphy) for mass-balance calculation of the subduction sulfur influx [see L338-342, Page 13]. In addition, we described the calculations and calibration procedure briefly of subduction sulfur output estimates [see L380-391, Page 15]. In order to display the depth-dependent H_2O flux from van Keken et al. (2011), we added a new diagram (Fig. 6b) to help the reader to understand our calculation and show how the major sulfur and water releases coincide at sub-arc depths.

Your discussion is actually very short and only speculative, I would suggest transferring much of the modeling information in this section (see above). Speculations on future research paths can be placed in such a journal, but they have to be limited 5-10 lines at the very end of the paper. Why not discussing modelling results with greater and clearer details (and a re-organisation) in the discussion instead?

Response: We extensively re-structured both the Results and Discussion sections. See general response point #2 to the editor.

Your data should be compared to recent papers by the Evans group that you have cited. These have evidenced high ^{34}S in eclogites, and one of the hypotheses to explain this is that mantle serpentinite dehydration from below the eclogitic crust continues to greater depths. Looking at your Fig. S1, adding these data would considerably change the picture. After all, you provide data on only one locality and the availability of oxidized and high ^{34}S from serpentinites below may also change.

Response: We compared our data to those published by the group of Evans et al. on sulfides in serpentinites and eclogites in the Results. See response to the Comment #2 of “Quality of the text and presentation” of Reviewer #2.

In the new Fig. 3, we do not report the $\delta^{34}\text{S}$ data on the Alps, New Caledonia and Corsica published in Evans et al. (2014) and Crossley et al. (2018), because there is no bulk sulfur concentration published in these references, so they cannot be plotted on the diagram. In addition, those data are in-situ SIMS $\delta^{34}\text{S}$ results of sulfide, not bulk-rock sulfur isotope measurements as shown in this diagram.

Figures are large and complex but clear and comprehensible, minor modifications need to be undertaken, such as symbols in Fig. 2c, which variable shape can be misleading. Symbols must be defined in each figure caption as well.

Response: For re-organization of figures, see our general response point #3 to the editor

Line by line comments of the main text and figures

Line 32-33: does this observation imply that sulfate transported by the slab is low or that it was lost during devolatilization?

Response: We rewrote this sentence to make it clearer [L36-37, Page 2]:

“Textual and thermodynamic evidence indicates the predominance of reduced sulfur species in slab fluids...”

Line 47-50: here you need to state that this putative high- ^{34}S addition is under the form of sulfate. Otherwise you wouldn't have such high S contents (up to 3000 ppm), which correspond to the S content at sulfate saturation in andesitic melts (Botcharnikov et al. 2011).

Response: Botcharnikov et al. (2011) indeed reported a S content of 3000 ppm at sulfate saturation in andesitic melts at 1050 °C, and 1500 ppm [S] at FeS saturation at lower $f\text{O}_2$. However, in basaltic melts, [S] at sulfate saturation reaches 6000 ppm and 3000 ppm at FeS saturation at lower $f\text{O}_2$. Sulfate solubility in melts is strongly dependent on temperature, and the maximum dissolved S in basaltic melt under oxidizing conditions is ~1.5 wt% S at 1300 °C and 1 GPa pressure (see review of Wallace and Edmonds, 2011). Considering the high $f\text{O}_2$ of many arc magmas, sulfur in arc melts is likely in the form of sulfate. We added “sulfate” to the sentence [see L55, Page 3].

Line 50-51: there are no such alternative models, this is actually the kind of model you propose in this MS. This statement (with no references) does not describe the following lines; i.e. suggesting that degassing can influence ^{34}S does not mean that (oxidized) S hasn't been added from the slab.

Degassing can increase ^{34}S , true, but first, the Mandeville's inclusions contain 300 ppm S at most. More recent papers by Fiege et al. (2014, 2015) have shown that the fractionation factors between melt and fluid can be also highly dependent on redox state, especially above the sulfide/sulfate transition at FMQ+1. Producing high ^{34}S in melts can be done at these conditions, which is actually very close to what is expected for primitive arc compositions (e.g. peridotites, Bénard et al. 2018 in Chem Geol).

The high sulfur in many arc inclusions has to come from somewhere, and undegassed (sulfate-saturated), high- ^{34}S and S (3000 ppm or more) sub-arc mantle inclusions have trace element spikes typical of slab agents (e.g. Bénard et al. 2018 in Nature Comm).

The real challenge of this research is somehow twisted with lines 50-54 coz ref. 10 says there is no sulfate in primary arc magmas, whereas only sulfate-present degassing can produce increases in ^{34}S significant enough to reproduce arc inclusions. If it is not oxidized, this means that ^{34}S is in the source as part of a recycled component or produced by the melting process.

Response: We appreciate the reviewer's clarification of the current debate about what causes the positive $\delta^{34}\text{S}$ values observed in arc volcanic rocks and melt inclusions. The fractionation factors between melt and fluid are highly dependent on the redox state. Degassing at oxidizing conditions (>FMQ+1.8) causes melts to get heavier in $\delta^{34}\text{S}$, whereas at intermediate to reducing conditions melts get lighter (Fiege et al., 2014). In general, the sulfur isotope fractionation caused by degassing is small (about +3.7‰ in reduced systems and -1.5‰ in oxidized systems) (Fiege et al., 2015). We do agree with the reviewer that degassing is not the debate focus of subduction zone sulfur cycles, so we removed this part as we described above. We focus on the current debate regarding whether there is a slab contribution of sulfate to the mantle wedge to define the scientific problem. We rewrote this part of the first two paragraphs of the main text [see L49-68, Page 3].

Line 53-54: same as above, this is not the debate that is currently in the community. In a Nature paper you need to carefully lay out the scientific problem, please re-write this paragraph accordingly.

Response: We thank the reviewer for helping us to sharpen the scientific problem. We rewrote these sentences. See the response above.

Line 65-69: are these pseudotachylite veins? Are they quenched? If not, you are probably overstating the importance of HP veins in directly reflecting the nature of the agents from which they formed (see overall comments).

Response: No, these veins are not pseudotachylite veins and are not quenched melts. They are formed from a free fluid phase flowing in fracture-like structure within rocks. They generally represent the fossil fluid activity and record critical information of fluid compositions and characteristics. See detailed response to the general comments.

Line 73-74: 'unparalleled' is really daring here, earlier studies have also looked at HP rocks and their sulfides...

Response: We replaced this word by "exceptional" [see L87, Page 4].

Line 77: what do you mean exactly by unaltered, that the sulfides present fresh primary textures? For some people, serpentinization means alteration, so you have to be more specific here, or alternatively, you could remove this word to later evidence precisely the degree of alteration of each phase in your petrographic description.

Response: We removed the word "unaltered" from the text.

Line 77-92: this part is well-written and clear in terms of ^{34}S results, but on the contrary to the following one on veins, it provides very little background information on samples. Yet, these can be really important at this stage to understand your objectives with these analyses. I am not talking about details and repeating what has been already published in ref. 27, but rather, I would have expected a short explanation of field relationships, and more importantly, thermobarometric estimates for metabasites, metapelites and serpentinites. How the reader is supposed to understand what these numbers mean if no information is provided about the devolatilization sequence?

Response: We added a “Sample Background” subsection. See our general response point #1 to editor.

Line 88: well, in that case this is not really suitable for a Nature paper. Furthermore, this is not entirely true coz Evans et al. (2014) have reported high ^{34}S in eclogites...

Response: See our response to the “Novelty and relevance” point raised at the beginning of the comment of Reviewer #2.

Evans et al. (2014) have reported in-situ positive $\delta^{34}\text{S}$ compositions of sulfides in eclogites from the Alps and New Caledonia. They interpret pyrite with $\delta^{34}\text{S}$ values around 7‰ as a result of pre-subduction hydrothermal alteration. But “...the sulfur may have moved, dissolved in a fluid, from one location to another during subduction zone metamorphism.” We agree with this interpretation, that the 7‰ sulfur may be derived from dehydration of slab serpentinite beneath subducted oceanic crust, since subducted serpentinites likely have positive $\delta^{34}\text{S}$ compositions (e.g., Alt et al., 2012; Crossley et al., 2018). We discuss these positive $\delta^{34}\text{S}$ compositions in eclogites when we interpret $\delta^{34}\text{S}$ sources [see L273-278, Page 11].

Line 112-114: this rather looks like interpretation, which must be transferred to the very beginning of the discussion.

Response: We moved the interpretation to the Discussion [see L382-385, Page 15].

Line 115-117: thermobarometric estimates should be specified for these rocks as well.

Response: In the new added “Sample Background” subsection, we provide the thermobarometric estimates for these vein samples [see L113-119, Page 5].

Line 146: an example of why adding more petrological background would be important, as these P-T values have to be checked in comparison with the formation conditions of all your samples. If the P-T conditions are the same for all samples (as mentioned very shortly, hidden to be honest, in the supplements), this has to be specified way more clearly as well.

Response: We added a “Sample Background” subsection. See our general response point #1 to editor.

Line 150-154: again, this means the results are not that novel.

Response: The reviewer may have a misunderstanding due to unclear wording on our part. We show that our measured $\delta^{34}\text{S}$ results of metapelites (negative) are consistent with their protolith marine sedimentary sediments (also negative $\delta^{34}\text{S}$ with comparable values), indicating that there is no large sulfur isotope fractionation during HP metamorphism for sediments. Thus, the observed vein pyrite with negative $\delta^{34}\text{S}$ (-8‰) likely represents the fluids derived from metasediments. We rewrote this sentence to make it clearer [see L261-267, Pages 10-11].

Line 158-160: where and how is this interpreted? In your paper? Based on the literature using other geochemical proxies? Please re-write.

Response: Here we interpret that our sample (Vein_1) records the fluid $\delta^{34}\text{S}$ compositions derived from oceanic crust, after a fully considered discussion. We rewrote the entire paragraph to make it clearer [see L268-284, Page 11].

Line 168-170: I didn't get the full logic here, please re-write and give more details. Do you mean that residual Co is stored in serpentinite sulfides after SO_4 flushing, and that it is thus normal to not have in in great quantities in the metasomatic pyrite? If so, please re-write accordingly.

Response: We deleted this sentence since it may cause a misunderstanding, and may not be robust evidence for tracing the fluid source [see L331-332, Page 11]

Line 173-174: can you be more specific and show what you are talking about, instead of just stating that things are consistent with your interpretation? Again, a preliminary petrological background part would be useful here, as it would prevent the appearance of new petrological observations (for the reader at least) in an odd manner in the results.

Response: We added text in the useful message in the "Sample Background" to describe the late-stage fractures surrounding pyrite grains [see L128-132, Pages 5-6]. We argue that the negative $\delta^{34}\text{S}$ (-5‰) of thin rims on vein pyrite (Figs. 2n and 5d) with the sharply increasing Co concentrations is related to this late-stage modification [see L294-297, Page 12].

Line 195: ok, this is where your data are important, crucial to understand the rationale of your study, please outline this way more clearly here and elsewhere in the MS when re-organizing it.

Response: We rewrote these sentences to make our statement clear. First, we introduce how the $\delta^{34}\text{S}$ of slab fluids can be calculated. Second, we calculate the net $\delta^{34}\text{S}$ value of fluids at 70–100 km. Then we calibrate this value considering the effects of isotope exchange during fluid-rock interaction. Lastly, we calculate the net $\delta^{34}\text{S}$ value of fluids at 30–230 km. See paragraph in main text [L380-394, Page 15].

Line 209-245: this is not really a discussion and rather a list of speculations. Since a model cannot really be considered as the central result of your study (these are the measurements right?), I would suggest transferring the models in the discussion.

The speculations are fine for the most part, but it should be condensed to the minimum and not constitute the core of what is called 'discussion'.

However, some aspects of your speculations have been heavily contradicted in the literature, such as the potential of H_2O as an oxidizer...

Response: We re-structured the Results and Discussion sections. See general response point #2 to the editor. Regarding H_2O alone as an oxidizer, see the response to Comment Line 188-189 of Reviewer #1.

Line 219-220: this observation should appear way before this part, such as a preliminary background data part (see above). Otherwise, this is disorganized.

Response: We moved this sentence to the beginning of the Discussion [see L306-308, Page 18].

Line 560-561 and 570-571: these statements are obscure, it is impossible to make the link between what is shown in the figure and these conclusions without explanations. Wouldn't it be better to put clearer sentences but only in the discussion part?

Response: We deleted these sentences from the figure captions, and put reworded versions in the Discussion.

Line 595-603: would be nice to see where your samples are in this diagram, in particular what they do represent in terms of P-T conditions and S cycle. This would nicely complement the background data text part I suggest.

Response: We added red stars into the diagram to show where our samples formed. See new Fig. 9.

Line by line comments of the supplements

Line 53-70: why not reporting the $\delta^{34}\text{S}$ data of Evans et al. recent papers on the Alps, New Caledonia and Corsica? Another question is what are the rocks that are not shaded in grey? Are these all chlorite harzburgite formed from antigorite breakdown?

Response: We do not report these $\delta^{34}\text{S}$ data on the Alps, New Caledonia and Corsica in the diagram (now as Fig. 3) because there is no bulk sulfur concentration published in these references, so they cannot be plotted on the diagram. See response to the Comment #7 of “Quality of the text and presentation” of Reviewer #2.

Shaded area in grey refers to $\delta^{34}\text{S}$ values of metabasites, which is used to compare with mantle value.

As published by Alt et al., 2012, all chlorite harzburgites are formed from antigorite breakdown.

Line 95: what do you mean by ‘Fe-saturated’ in this context? Fe-rich?

Response: Yes, we replaced “Fe-saturated” with “Fe-rich”.

Reviewers' comments:

Reviewer #1 (Remarks to the Author):

I have now read the detailed point-by-point response to the Reviewer comments, as well as the revised manuscript and supplementary material. After doing so, I believe the authors have adequately addressed my concerns and critiques. I recommend the paper be published in Nature Communications. Thanks for the opportunity to review this work; I look forward to seeing the manuscript published.

Reviewer #2 (Remarks to the Author):

--Review of NCOMMS-19-19518A

This is the second time I review this MS. The authors did a serious work to improve some key aspects of the presentation (both text and figures) and I think this MS can ultimately be published in Nat. Comm because the data are of quality. I am generally convinced by their rebuttal letter and I am very satisfied by the presence of the background part, which really clarifies the objectives and the implications of this study.

However, I still have issues with how the debate in the literature on arc fO_2 is laid out in the introduction and how the implications of the results are put forward. This regards now the dichotomy of observations between intra-oceanic and immature continental arcs vs. mature continental arcs. In particular, maturity refers here to crustal thickness and modes of mantle melting and the dichotomy is typically illustrated by recent reports on rocks from Western vs. Eastern Pacific arcs.

Since I did a lengthy review in the first place, and that most of my and the other reviewer's detailed comments were taken into account, I only treat this issue, which really problematic all throughout the MS.

Line 53-58: here you compare observations from Western Pacific and Eastern Pacific arcs, this does not make sense. The dichotomy regarding the modes of mantle wedge melting between Western and Eastern Pacific arcs has been demonstrated using stable isotopes (Foden et al. 2018 in EPSL, not cited; this must be corrected). Western arcs record flux melting, and are therefore more prone to be influenced by an oxidizing agent derived from the slab, whereas Eastern arcs record decompression melting. So the debate on whether slab-derived agents can oxidize the mantle wedge or not cannot be presented by simply comparing observations from the two systems, this is misleading. What you can do here is keep the references as such but outline clearly that since the two systems cannot be compared, the solution to the debate remains unclear and new insights from HP rocks must be provided. But as written below, SO_4 has been found in primary arc magmas from Western Pacific arcs.

Without any surprise, sub-arc mantle portions presenting an elevated fO_2 between FMQ and FMQ+1 have been only identified in intra-oceanic or rare continental arcs with relatively thin crust, mainly in the Western Pacific (check Bénard et al. 2018 in Chemical Geology for the latest data compilation, not cited; this must be corrected). Instead, mantle peridotites from the subcontinental lithosphere at Eastern Pacific SZ have been less recognized as oxidized (Kilgore et al. 2018 in G3 and earlier references therein, such as Canil et al. 1990 in CMP vs Brandon and Draper 1996 in GCA). In fact, all observations point to very different systems that cannot be compared readily.

Furthermore, you wrongly twist the literature in this part again by stating 'arc settings have been attributed to the addition of slab-derived sulfate...Alternatively, deep arc cumulates with mantle-like $\delta^{34}S$ values indicate that there is no sulfate in primary arc magmas, and the positive $\delta^{34}S$ signature of arc lavas results from crustal assimilation¹⁰.' Some of the references you cite here (e.g. Bénard et al. 2018 in Nat Comm) have directly linked SO_4 (XANES) with elevated $\delta^{34}S$ (SIMS) from two Western Pacific arcs; since these are direct measurements, there is no doubt that

there is SO₄ in these primary arc magmas. So the real debate is to which extent different modes of mantle wedge melting allow recycling in primary arc magmas and your introduction must reflect the real nature of the results in the paper you cite.

Finally, the view that arc magmas are not oxidized from V and Zn systematics has been recently challenged, based on the T-dependence of their partitioning coefficients (Wang et al. 2019 in JGR, not cited; this must be corrected). Talks at the Goldschmidt 2019 Conference have outlined this effect in experiments done by other groups. So you also have to tone down the interpretation given by Lee et al. works, which heavily rely on EP arc cumulates or V and Zn systematics, which do not take into account the effects of different mantle melting modes between MOR and arcs. Otherwise, the introduction in the paper is not an accurate reflection of the current literature.

Line 63-67: this part cannot be left as such again. The papers you cite have made inferences, which must be summarized shortly but accurately. Statements like 'people have studied this and (we think/judge) this is inconclusive' are unacceptable, even with tight space constraints. The right way would be to have perhaps less citations with the same message but described with a statement such as 'some studies have outlined this.... whereas other studies have suggested that... Therefore, the real interplay between this and that remains unclear and new data on HP rocks... etc ...'

Line 63-67 and 77-82: here again, you miss important recent reports, which have shown S-driven oxidation can occur in subduction HP rocks (Rielli et al. 2017 in GPL, not cited; this must be corrected)... As I said in my first review, oxidation by SO₄ is a transient process and it is not because you don't find abundant sulfates in the residues of redox reactions that it hasn't been transported from the slab by the fluids and that it isn't observed in primary arc magmas in the end (Bénard et al., Nat Comm). Rielli et al. have observed small amounts of Ba sulfate and typical sulfides formed in oxidizing environment with increased Fe³⁺/Fe^{tot} in grt. Of course there is little sulfate remaining in these HP rocks as well, simply because it has been consumed by the redox reaction. Therefore, even some observations from HP rocks have outlined oxidation by S, so that it must be reflected in your MS accurately.

Lines 404-405, 417-435: here are many other points that justify being much more specific here to avoid blanket statements that are supposed to broaden the impact of the study. Instead, this dilutes the message and the real relevance/impact of the results. The authors must clearly distinguish between continental and oceanic arc settings or Western and Eastern Pacific arcs (see above). They also need to avoid overstating the implications of their findings by using the sole word 'arc' or by referring to the debate based on V-Zn systematics as an alternative option to observations of SO₄ in primary arc magmas (Bénard et al. 2018 in Nat Comm, see above); this is an inappropriate reflection of the real challenge the community is facing. This must be thoroughly corrected all throughout the MS.

Line 421: the processes of H₂ incorporation in opx is inconsistent with the high fO₂ in residues (Bénard et al. 2018, Chem Geol) and SO₄ in primary arc melts (Bénard et al., Nat Comm) from the SAME ROCKS. This must be corrected.

Line 430-432: again, a comparison between very different systems (refs 8 and 10) is useless and misleading for the scientific community ... You need to clearly define all aspects of the real debate FROM THE BEGINNING in your MS to avoid this (see above) ...

Line 432-435: this must be revised since slab agents at depths of kimberlite generation cannot be the same as those at depths of arc magma generation. The only thing you can say here is that negative δ³⁴S such as those you identify can be transferred way beyond the zones of arc magma generation! Furthermore, no mantle melting process capable of fractionating significantly S isotopes has been evidenced so far (check Labidi et al. works), so either you propose something clear and substantiated, or rather, I suggest you delete this part.

Lines 435 onwards: if it is nice to finally see that the authors acknowledge the complexity of different types of subduction zone systems, the writing is inappropriate here. The authors first claim 'our results surely demonstrate S is definitely NOT the oxidizer in arcs in general' to finally tone down to say 'in fact, this is very complicated and might be highly variable from one arc to another' ...

I strongly disagree with such a communication method, and I would like to see exactly the opposite in a scientific paper. This means something like 'the systems are complex and may vary from one setting to another, as shown by global data (e.g. redox-sensitive Fe isotopes), furthermore earlier S-in-arc data reports show that some HP rocks, particularly serpentinites, and some primary arc magmas DO CONTAIN SO₄, whereas other types of HP rocks do not (introduction). Indeed, SO₄-flushing might not be the case for all arcs, as our data suggest that little SO₄ is transferred to subarc depths FOR THE ROCKS STUDIED (last part of discussion). This may explain the variability of fO₂, d₃₄ ... observed among different types of arcs...' This must be rewritten as such all throughout the MS.

(Note: Reviewer comments are listed in blue, and responses to reviewer comments are in black. Line and page numbers in [brackets] refer to those in revised Manuscript_R2 with track markers)

Response to Anonymous Reviewer #2:

Reviewer #2 (Remarks to the Author):

Review of NCOMMS-19-19518A

This is the second time I review this MS. The authors did a serious work to improve some key aspects of the presentation (both text and figures) and I think this MS can ultimately be published in Nat. Comm because the data are of quality. I am generally convinced by their rebuttal letter and I am very satisfied by the presence of the background part, which really clarifies the objectives and the implications of this study. However, I still have issues with how the debate in the literature on arc fO_2 is laid out in the introduction and how the implications of the results are put forward. This regards now the dichotomy of observations between intra-oceanic and immature continental arcs vs. mature continental arcs. In particular, maturity refers here to crustal thickness and modes of mantle melting and the dichotomy is typically illustrated by recent reports on rocks from Western vs. Eastern Pacific arcs.

Since I did a lengthy review in the first place, and that most of my and the other reviewer's detailed comments were taken into account, I only treat this issue, which really problematic all throughout the MS.

Response: We appreciate the reviewer's positive comments on our revised manuscript. We modified the Introduction and Discussion sections accordingly, as suggested by the reviewer. The issue that the reviewer raises is addressed below.

Line 53-58: here you compare observations from Western Pacific and Eastern Pacific arcs, this does not make sense. The dichotomy regarding the modes of mantle wedge melting between Western and Eastern Pacific arcs has been demonstrated using stable isotopes (Foden et al. 2018 in EPSL, not cited; this must be corrected). Western arcs record flux melting, and are therefore more prone to be influenced by an oxidizing agent derived from the slab, whereas Eastern arcs record decompression melting. So the debate on whether slab-derived agents can oxidize the mantle wedge or not cannot be presented by simply comparing observations from the two systems, this is misleading. What you can do here is keep the references as such but outline clearly that since the two systems cannot be compared, the solution to the debate remains unclear and new insights from HP rocks must be provided. But as written below, SO_4 has been found in primary arc magmas from Western Pacific arcs.

Without any surprise, sub-arc mantle portions presenting an elevated fO_2 between FMQ and FMQ+1 have been only identified in intra-oceanic or rare continental arcs with relatively thin crust, mainly in the Western Pacific (check Bénard et al. 2018 in Chemical Geology for the latest data compilation, not cited; this must be corrected). Instead, mantle peridotites from the subcontinental lithosphere at Eastern Pacific SZ have been less recognized as oxidized (Kilgore et al. 2018 in G3 and earlier references therein, such as Canil et al. 1990 in CMP vs Brandon and Draper 1996 in GCA). In fact, all observations point to very different systems that cannot be compared readily.

Response: We appreciate the reviewer's perspective on the current debate about oxidization states of different arc settings. We acknowledge that the Western Pacific arcs are a result of flux melting, which may show a clear record of metasomatism of slab fluids. However, we do not understand why the Eastern Pacific arcs are supposed to be related to a decompression melting mode. Decompression melting commonly occurs at divergent plate boundaries, like in the upwelling part of a mantle convection cell (mid-ocean ridges), at a mantle plume (hotspots), or during continental lithospheric extension (rifts). Solely Fe isotope data of arc rocks cannot be treated as a solid evidence of decompression melting beneath Eastern Pacific arcs, in the absence of geophysical and other geological evidences. But regardless, we broadly agree that differences in such geologic factors as melting mode and arc maturity make direct comparisons difficult. We have outlined the salient characteristics of different arc systems and noted the challenges of direct comparison in the Discussion [see L324-331, Page 13]. The suggested Foden et al. reference is now cited. We only added the "(e.g., Western Pacific)" and "(e.g., Eastern Pacific)" into two

sentences of Introduction [see L46-52, Page 3], but did not go into great detail as our study is focusing on slab rocks rather than arc rocks.

Furthermore, you wrongly twist the literature in this part again by stating ‘arc settings have been attributed to the addition of slab-derived sulfate...Alternatively, deep arc cumulates with mantle-like $\delta^{34}\text{S}$ values indicate that there is no sulfate in primary arc magmas, and the positive $\delta^{34}\text{S}$ signature of arc lavas results from crustal assimilation¹⁰.’ Some of the references you cite here (e.g. Bénard et al. 2018 in Nat Comm) have directly linked SO_4 (XANES) with elevated $\delta^{34}\text{S}$ (SIMS) from two Western Pacific arcs; since these are direct measurements, there is no doubt that there is SO_4 in these primary arc magmas. So the real debate is to which extent different modes of mantle wedge melting allow recycling in primary arc magmas and your introduction must reflect the real nature of the results in the paper you cite.

Response: We changed the “indicate that there is no sulfate in primary arc magmas” to “suggest more limited slab-derived sulfate contributions to arc lavas” [see L51-52, Page 3].

Finally, the view that arc magmas are not oxidized from V and Zn systematics has been recently challenged, based on the T-dependence of their partitioning coefficients (Wang et al. 2019 in JGR, not cited; this must be corrected). Talks at the Goldschmidt 2019 Conference have outlined this effect in experiments done by other groups. So you also have to tone down the interpretation given by Lee et al. works, which heavily rely on EP arc cumulates or V and Zn systematics, which do not take into account the effects of different mantle melting modes between MOR and arcs. Otherwise, the introduction in the paper is not an accurate reflection of the current literature.

Response: We respectively disagree with the reviewer, as Lee et al.’s study cited here is not based on V and Zn systematics. Nonetheless, considering recent controversy regarding interpretation of V and Zn behavior in the literature, we deleted the discussion and citation related to conclusions based on V and Zn systematics [see L361, Page 14; L530-531, Page 19]. Our study does not utilize V or Zn and thus there is no loss of impact or change in our conclusions.

Line 63-67: this part cannot be left as such again. The papers you cite have made inferences, which must be summarized shortly but accurately. Statements like ‘people have studied this and (we think/judge) this is inconclusive’ are unacceptable, even with tight space constraints. The right way would be to have perhaps less citations with the same message but described with a statement such as ‘some studies have outlined this... whereas other studies have suggested that... Therefore, the real interplay between this and that remains unclear and new data on HP rocks... etc ...’

Response: We rewrote this sentence and summarized the findings concisely in the style suggested by the reviewer [see L57-62, Page 3].

Line 63-67 and 77-82: here again, you miss important recent reports, which have shown S-driven oxidation can occur in subduction HP rocks (Rielli et al. 2017 in GPL, not cited; this must be corrected)... As I said in my first review, oxidation by SO_4 is a transient process and it is not because you don’t find abundant sulfates in the residues of redox reactions that it hasn’t been transported from the slab by the fluids and that it isn’t observed in primary arc magmas in the end (Bénard et al., Nat Comm). Rielli et al. have observed small amounts of Ba sulfate and typical sulfides formed in oxidizing environment with increased $\text{Fe}^{3+}/\text{Fe}^{\text{tot}}$ in grt. Of course there is little sulfate remaining in these HP rocks as well, simply because it has been consumed by the redox reaction. Therefore, even some observations from HP rocks have outlined oxidation by S, so that it must be reflected in your MS accurately.

Response: It is a bit surprising that yet another paper of the same group of authors must be cited. For most of the previously mentioned papers it was arguable to add citations but still acceptable, as rapidly evolving science always leads to different points of view. We appreciate that this kind of intense discussion contributes to scientific advances.

However, Rielli et al. (2017, GPL) do not report on subduction zone HP rocks. Their rocks belong to the Western Gneiss Complex of Norway which was part of the passive continental margin of Baltica subducted during the Caledonian continental collision. Rielli et al. studied mantle peridotites, which have been thought to be metasomatized by fluids/melts, with probable slab signatures, if they belonged to the upper plate at all (debated in the literature). Note that the origin of metasomatism is also still in debate, but most likely occurred from the fluids/melts of the surrounding rocks which are of continental origin, see description in Rielli et al. (2017 GPL). Since their paper concerns mantle peridotite, and because they are from a locality corresponding to a geological setting that has nothing to do with Pacific arcs, we do not add this information since we focus on subduction zone HP rocks.

Regarding whether there is SO₄ in slab fluids, our argument has already been provided in our last rebuttal letter to response Comment #2 of “Scientific merit” of Reviewer #2.

Lines 404-405, 417-435: here are many other points that justify being much more specific here to avoid blanket statements that are supposed to broaden the impact of the study. Instead, this dilutes the message and the real relevance/impact of the results. The authors must clearly distinguish between continental and oceanic arc settings or Western and Eastern Pacific arcs (see above). They also need to avoid overstating the implications of their findings by using the sole word ‘arc’ or by referring to the debate based on V-Zn systematics as an alternative option to observations of SO₄ in primary arc magmas (Bénard et al. 2018 in Nat Comm, see above); this is an inappropriate reflection of the real challenge the community is facing. This must be thoroughly corrected all throughout the MS.

Response: As suggested by the reviewer, here we distinguish and describe the Western and Eastern Pacific arcs, and state that the thermal regime of the Tianshan rocks we study corresponds more closely to the Western Pacific arc style [see L324-333, Page 13]. As noted above, we deleted the discussion and citation related to conclusions based on V and Zn systematics [see L361, Page 14; L530-531, Page 19]. In addition, we endeavored to make our arguments more concise, see below.

Line 421: the processes of H₂ incorporation in opx is inconsistent with the high fO₂ in residues (Bénard et al. 2018, Chem Geol) and SO₄ in primary arc melts (Bénard et al., Nat Comm) from the SAME ROCKS. This must be corrected.

Response: Yes, Bénard et al. (2018, Che. Geol.) and Tollan & Hermann (2019, Nat. Geosci.) studied the same sets of mantle peridotite xenoliths from the West Bismarck arc. Two types of mantle peridotites are identified: *Type_1* has well-equilibrated protogranular texture with low fO₂ (FMQ-0.8 to FMQ+0.2), whereas *Type_2* shows reaction textures (recording rock-melt interaction?) with high fO₂ (FMQ+1 to FMQ+1.5). Bénard et al. (2018, Che. Geol.) interpreted that *Type_1* represent residual peridotites by first-stage flux melting and *Type_2* as residual peridotites by second-stage decompression melting. Tollan & Hermann (2019, Nat. Geosci.) argued that the primary arc magmas are probably reduced. The water dissociation and H₂ incorporation in surrounding Opx produces gradually oxidized melts and *Type_1* peridotite (ambient mantle) with low fO₂, whereas the immediate peridotites along the melt channel forms high fO₂ *Type_2* peridotite by interaction with passing, already oxidized melts (may be SO₄-bearing). The formation mechanism of the Ritter peridotites is still in debate, but Tollan & Hermann’s new work is indeed internally consistent with the observations. We keep the sentence here.

Line 430-432: again, a comparison between very different systems (refs 8 and 10) is useless and misleading for the scientific community ... You need to clearly define all aspects of the real debate FROM THE BEGINNING in your MS to avoid this (see above) ...

Response: We deleted the comparison described here [see L345-348, Page 14]. As discussed above, we have rewritten the discussion part [see L324-331, Page 13] to help clarify the debate.

Line 432-435: this must be revised since slab agents at depths of kimberlite generation cannot be the same as those at depths of arc magma generation. The only thing you can say here is that negative $\delta^{34}\text{S}$ such as those you identify can be transferred way beyond the zones of arc magma generation! Furthermore, no mantle melting process capable of fractionating significantly S isotopes has been evidenced so far (check Labidi et al. works), so either you propose something clear and substantiated, or rather, I suggest you delete this part.

Response: Deleted [see L348-350, Page 14].

Lines 435 onwards: if it is nice to finally see that the authors acknowledge the complexity of different types of subduction zone systems, the writing is inappropriate here. The authors first claim 'our results surely demonstrate S is definitely NOT the oxidizer in arcs in general' to finally tone down to say 'in fact, this is very complicated and might be highly variable from one arc to another' ...

I strongly disagree with such a communication method, and I would like to see exactly the opposite in a scientific paper. This means something like 'the systems are complex and may vary from one setting to another, as shown by global data (e.g. redox-sensitive Fe isotopes), furthermore earlier S-in-arc data reports show that some HP rocks, particularly serpentinites, and some primary arc magmas DO CONTAIN SO_4 , whereas other types of HP rocks do not (introduction). Indeed, SO_4 -flushing might not be the case for all arcs, as our data suggest that little SO_4 is transferred to subarc depths FOR THE ROCKS STUDIED (last part of discussion). This may explain the variability of $f\text{O}_2$, $\delta^{34}\text{S}$... observed among different types of arcs...' This must be rewritten as such all throughout the MS.

Response: There might be a misunderstanding here. We did not write the quoted sentences in our manuscript "*our results surely demonstrate S is definitely NOT the oxidizer in arcs in general*" and "*in fact, this is very complicated and might be highly variable from one arc to another*". But regardless, we deleted this part since it may cause misunderstanding. In addition, we appreciate the reviewer's enthusiasm, certainly, understanding the variability of $f\text{O}_2$ and $\delta^{34}\text{S}$ values observed among different types of arcs is of great importance. But as we have emphasized, our work focuses on slab rocks. Too much discussion from the viewpoint of arcs is beyond the scope of this study.

REVIEWERS' COMMENTS:

Reviewer #2 (Remarks to the Author):

Review of NCOMMS-19-19518B.

The authors did the last text re-working steps that were needed to improve the message and avoid the distracting and bold statements. The results are now clearly put into the specific context of the Southern Tianshan Ocean subduction. The MS is now ready for publication, with minor adjustments.

I understand that the authors don't want to cite Rieli et al. 2017, if there is a debate on the origin of these rocks (even though these samples contain SO₄-species, which doesn't go well with their statements that SO₄ is rare in HP rocks near or in the slab). However, regarding the answer of the authors to my comment on H incorporation into opx to oxidize arc magmas (Tollan and Hermann 19) vs. the earlier report by Benard et al. 2018, the authors should be careful when citing these papers here. All 'residual' rocks reported by Benard et al. 2018 have protogranular texture, are free of melt-rock reaction and present fO₂ ranging from FMQ-0.5 to FMQ+1.5 (blue symbols in their figures). Scans of these rocks are available in an earlier paper by Benard et al. 2017 in GCA, so this is not disputable. Only 3 dunitic channels and one harzburgite with SO₄-bearing Mis can be defined as 'melt-percolated' in the samples reported by Benard et al. 2018 (white symbols). So the fact that some residual Ritter rocks are oxidized is not disputed, as Tollan and Hermann have just used the fO₂ data previously reported by Benard et al, but only for the reduced residues...

H incorporation in opx might play a role I certainly agree, however, Tollan and Hermann have not reported IR data for oxidized residues from Ritter, and therefore, the mechanism you propose for enriching melt in ³⁴S during melting does not rely on existing data (see below).

That post-melting metasomatism tends to increase fO₂ in the mantle lithosphere is definitely not new and this occurs in all tectonic settings. For the case of Ritter samples, increasing Fe³⁺/Fetot in spinel used to calculate fO₂ is related to the presence of SO₄ in percolating slab melts (the other paper by Benard et al. 2018 you have cited). So in the end, even the papers you cite show the contrary of what you propose in terms of oxidation and ³⁴S enrichment during melting.

In any cases, as I have already said, the quality of the data the authors report is sufficiently important to justify publication. I believe they have the potential to contribute to the debate about where SO₄ is liberated from the slab and why some mantle wedge portions do reflect this process and others not.

Line numbers below refer to the article pdf file including track changes.

Line 29: textural

Line 46-47: here you need to add something like 'the presence of sulfate, ', for instance between '3000 μg/g)' and 'and positive...' and specify that these are 'mantle xenoliths'. The fact that SZ melt inclusions do contain SO₄ (i.e. it is not an interpretation of elevated ³⁴S), even in the subarc mantle where it is linked with increased Fe³⁺/Fetot in minerals, should be clearly said here.

Line 350-354: this part cannot be left as such, firstly for the reasons detailed above. Furthermore, the subarc mantle rocks have experienced 25-45% cumulated melting at 1-2 GPa and more than 1250°C (no or little residual cpx). So there is no chance you will leave residual sulfide (see latest data by Zhang and Hirschmann 16). As a matter of fact, these rocks contain no primary sulfide inclusions in olivine (the only phase that will be produced during hydrous mantle melting and that could shield sulfide). Furthermore if you oxidize during melting, you will fully destabilize immiscible sulfide melt for even lower P-T conditions than those in Zhang and Hirschmann 16, so your hypothetical process does not stand, both from the point of earlier data and self-consistency.

I suggest deleting this or just point that the question of how SO₄ and high d³⁴S occur in the source of arc magmas remain open.

(Note: Reviewer comments are listed in blue, and responses to reviewer comments are in black. Line and page numbers in [brackets] refer to those in revised Manuscript_R3 with track changes)

Response to Anonymous Reviewer #2:

Reviewer #2 (Remarks to the Author):
Review of NCOMMS-19-19518B.

The authors did the last text re-working steps that were needed to improve the message and avoid the distracting and bold statements. The results are now clearly put into the specific context of the Southern Tianshan Ocean subduction. The MS is now ready for publication, with minor adjustments.

Response: We appreciate the reviewer's positive comments on our revised manuscript.

I understand that the authors don't want to cite Rieli et al. 2017, if there is a debate on the origin of these rocks (even though these samples contain SO₄-species, which doesn't go well with their statements that SO₄ is rare in HP rocks near or in the slab).

Response: We appreciate the reviewer's understanding.

However, regarding the answer of the authors to my comment on H incorporation into opx to oxidize arc magmas (Tollan and Hermann 19) vs. the earlier report by Benard et al. 2018, the authors should be careful when citing these papers here. All 'residual' rocks reported by Benard et al. 2018 have protogranular texture, are free of melt-rock reaction and present fO₂ ranging from FMQ-0.5 to FMQ+1.5 (blue symbols in their figures). Scans of these rocks are available in an earlier paper by Benard et al. 2017 in GCA, so this is not disputable. Only 3 dunitic channels and one harzburgite with SO₄-bearing Mts can be defined as 'melt-percolated' in the samples reported by Benard et al. 2018 (white symbols). So the fact that some residual Ritter rocks are oxidized is not disputed, as Tollan and Hermann have just used the fO₂ data previously reported by Benard et al, but only for the reduced residues...

H incorporation in opx might play a role I certainly agree, however, Tollan and Hermann have not reported IR data for oxidized residues from Ritter, and therefore, the mechanism you propose for enriching melt in ³⁴S during melting does not rely on existing data (see below).

That post-melting metasomatism tends to increase fO₂ in the mantle lithosphere is definitely not new and this occurs in all tectonic settings. For the case of Ritter samples, increasing Fe³⁺/Fe^{tot} in spinel used to calculate fO₂ is related to the presence of SO₄ in percolating slab melts (the other paper by Benard et al. 2018 you have cited). So in the end, even the papers you cite show the contrary of what you propose in terms of oxidation and ³⁴S enrichment during melting.

Response: We greatly appreciate the suggestions and enthusiasm from the second reviewer. We are tackling deep sulfur cycling from the slab perspective. We are happy to present relevant implications of our data for the currently widely diverging views of arc magmas processes but have to leave the details to those studies that deal directly with arc samples. We now propose our argument based on our data, and leave it as an open question (see our response to the last comment) to help readers make their own assessments.

In any cases, as I have already said, the quality of the data the authors report is sufficiently important to justify publication. I believe they have the potential to contribute to the debate about where SO₄ is liberated from the slab and why some mantle wedge portions do reflect this process and others not.

Response: We appreciate the reviewer's positive comments on our revised manuscript.

Line numbers below refer to the article pdf file including track changes.

Line 29: textural

Response: Corrected. [see Line 29, Page 2]

Line 46-47: here you need to add something like ‘the presence of sulfate, ’, for instance between ‘3000 □g/g) and ‘and positive...’ and specify that these are ‘mantle xenoliths’. The fact that SZ melt inclusions do contain SO₄ (i.e. it is not an interpretation of elevated d₃₄S), even in the subarc mantle where it is linked with increased Fe³⁺/Fe²⁺ in minerals, should be clearly said here.

Response: Added. [see Line 49, Page 3]

Line 350-354: this part cannot be left as such, firstly for the reasons detailed above. Furthermore, the subarc mantle rocks have experienced 25-45% cumulated melting at 1-2 GPa and more than 1250°C (no or little residual cpx). So there is no chance you will leave residual sulfide (see latest data by Zhang and Hirschmann 16). As a matter of fact, these rocks contain no primary sulfide inclusions in olivine (the only phase that will be produced during hydrous mantle melting and that could shield sulfide). Furthermore if you oxidize during melting, you will fully destabilize immiscible sulfide melt for even lower P-T conditions than those in Zhang and Hirschmann 16, so your hypothetical process does not stand, both from the point of earlier data and self-consistency.

I suggest deleting this or just point that the question of how SO₄ and high d₃₄S occur in the source of arc magmas remain open.

Response: Yes. In principle, a mantle rock that has suffered >20% partial melting has no sulfide left. However, geological processes may be more complex. Previous studies have identified sulfides from mantle peridotites, mantle xenoliths, and as inclusions in diamond (see review in Kiseeva et al., 2017 Elements). In addition, our argument proposes sulfide as a possible product in the immediate mantle rock during melt-rock interaction process, not as a residual phase during partial melting. But as suggested by the reviewer, we added one sentence at the end of this paragraph to point out that this as an open question and requires further study. [see Lines 356-357, Page 14]